# METRA: Scalable Unsupervised RL with Metric-Aware Abstraction

**Seohong Park**[1]    **Oleh Rybkin**[1]    **Sergey Levine**[1]
[1]University of California, Berkeley
`seohong@berkeley.edu`

## Abstract

Unsupervised pre-training strategies have proven to be highly effective in natural language processing and computer vision. Likewise, unsupervised reinforcement learning (RL) holds the promise of discovering a variety of potentially useful behaviors that can accelerate the learning of a wide array of downstream tasks. Previous unsupervised RL approaches have mainly focused on pure exploration and mutual information skill learning. However, despite the previous attempts, making unsupervised RL truly scalable still remains a major open challenge: pure exploration approaches might struggle in complex environments with large state spaces, where covering every possible transition is infeasible, and mutual information skill learning approaches might completely fail to explore the environment due to the lack of incentives. To make unsupervised RL scalable to complex, high-dimensional environments, we propose a novel unsupervised RL objective, which we call **Metric-Aware Abstraction** (**METRA**). Our main idea is, instead of directly covering the entire state space, to only cover a compact latent space $\mathcal{Z}$ that is *metrically* connected to the state space $\mathcal{S}$ by temporal distances. By learning to move in every direction in the latent space, METRA obtains a tractable set of diverse behaviors that approximately cover the state space, being scalable to high-dimensional environments. Through our experiments in five locomotion and manipulation environments, we demonstrate that METRA can discover a variety of useful behaviors even in complex, pixel-based environments, being the **first** unsupervised RL method that discovers diverse locomotion behaviors in pixel-based Quadruped and Humanoid. Our code and videos are available at https://seohong.me/projects/metra/

## 1 Introduction

Unsupervised pre-training has proven transformative in domains from natural language processing to computer vision: contrastive representation learning (Chen et al., 2020) can acquire effective features from unlabeled images, and generative autoregressive pre-training (Brown et al., 2020) can enable language models that can be adapted to a plethora of downstream applications. If we could derive an equally scalable framework for unsupervised reinforcement learning (RL) that autonomously explores the space of possible behaviors, then we could enable general-purpose unsupervised pre-trained agents to serve as an effective foundation for efficiently learning a broad range of downstream tasks. Hence, our goal in this work is to propose a scalable unsupervised RL objective that encourages an agent to explore its environment and learn a breadth of potentially useful behaviors without any supervision.

While this formulation of unsupervised RL has been explored in a number of prior works, making fully unsupervised RL truly scalable still remains a major open challenge. Prior approaches to unsupervised RL can be categorized into two main groups: pure exploration methods (Burda et al., 2019; Pathak et al., 2019; Liu & Abbeel, 2021b; Mendonca et al., 2021; Rajeswar et al., 2023) and unsupervised skill discovery methods (Eysenbach et al., 2019a; Sharma et al., 2020; Laskin et al., 2022; Park et al., 2022). While these approaches have been shown to be effective in several unsupervised RL benchmarks (Mendonca et al., 2021; Laskin et al., 2021), it is not entirely clear whether such methods can indeed be scalable to complex environments with high intrinsic dimensionality. Pure exploration-based unsupervised RL approaches aim to either *completely* cover the entire state space (Burda et al., 2019; Liu & Abbeel, 2021b) or *fully* capture the transition dynamics of the Markov decision process (MDP) (Pathak et al., 2019; Sekar et al., 2020; Mendonca et al., 2021; Rajeswar et al., 2023). However, in complex environments with a large state space, it may be

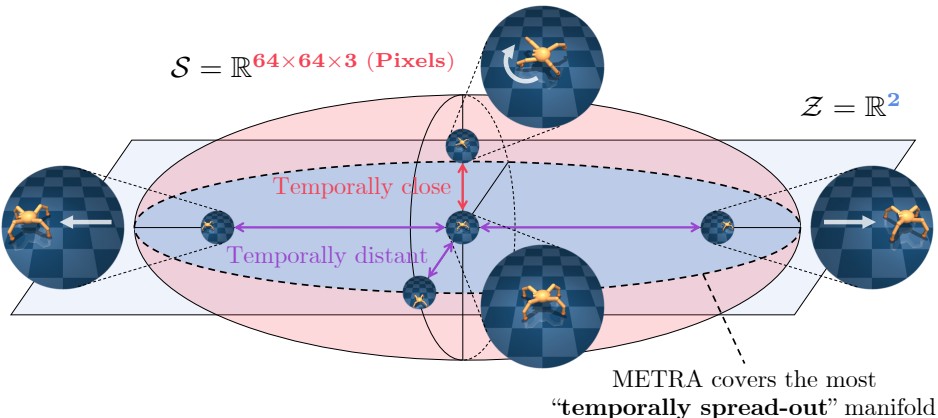

Figure 1: **Illustration of METRA.** Our main idea for scalable unsupervised RL is to cover only the most "important" low-dimensional subset of the state space, analogously to PCA. Specifically, METRA covers the most "temporally spread-out" (non-linear) manifold, which would lead to *approximate* coverage of the state space $\mathcal{S}$. In the example above, the two-dimensional $\mathcal{Z}$ space captures behaviors running in all directions, not necessarily covering every possible leg pose.

infeasible to attain either of these aims. In fact, we will show that these methods fail to cover the state space even in the state-based 29-dimensional MuJoCo Ant environment. On the other hand, unsupervised skill discovery methods aim to discover diverse, distinguishable behaviors, *e.g.*, by maximizing the mutual information between skills and states (Gregor et al., 2016; Eysenbach et al., 2019a). While these methods do learn behaviors that are mutually different, they either do not necessarily encourage exploration and thus often have limited state coverage in the complete absence of supervision (Eysenbach et al., 2019a; Sharma et al., 2020), or are not directly scalable to pixel-based control environments (Park et al., 2022; 2023b).

In this work, we aim to address these challenges and develop an unsupervised RL objective, which we call **Metric-Aware Abstraction** (**METRA**), that scales to complex, image-based environments with high intrinsic dimensionality. Our first main idea is to learn diverse behaviors that maximally cover *not* the original state space but a compact *latent metric space* defined by a mapping function $\phi : \mathcal{S} \rightarrow \mathcal{Z}$ with a metric $d$. Here, the latent state is connected by the state space by the metric $d$, which ensures that covering latent space leads to coverage of the state space. Now, the question becomes which metric to use. Previous metric-based skill learning methods mostly used the Euclidean distance (or its scaled variant) between two states (He et al., 2022; Park et al., 2022; 2023b). However, such state-based metrics are not directly applicable to complex, high-dimensional state space (*e.g.*, images). Our second main idea is therefore to use *temporal distances* (*i.e.*, the number of minimum environment steps between two states) as a metric for the latent space. Temporal distances are invariant to state representations and thus applicable to pixel-based environments as well. As a result, by maximizing coverage in the compact latent space, we can acquire diverse behaviors that approximately cover the entire state space, being scalable to high-dimensional, complex environments (Figure 1).

Through our experiments on five state-based and pixel-based continuous control environments, we demonstrate that our method learns diverse, useful behaviors, as well as a compact latent space that can be used to solve various downstream tasks in a zero-shot manner, outperforming previous unsupervised RL methods. To the best of our knowledge, METRA is the **first** unsupervised RL method that demonstrates the discovery of diverse locomotion behaviors in **pixel-based** Quadruped and Humanoid environments.

## 2 WHY MIGHT PREVIOUS UNSUPERVISED RL METHODS FAIL TO SCALE?

The goal of unsupervised RL is to acquire useful knowledge, such as policies, world models, or exploratory data, by interacting with the environment in an unsupervised manner (*i.e.*, without tasks or reward functions). Typically, this knowledge is then leveraged to solve downstream tasks more efficiently. Prior work in unsupervised RL can be categorized into two main groups: pure exploration methods and unsupervised skill discovery methods. Pure exploration methods aim to cover the entire state space or fully capture the environment dynamics. They encourage exploration by maximizing uncertainty (Pathak et al., 2017; Shyam et al., 2019; Burda et al., 2019; Pathak et al., 2019; Sekar et al., 2020; Mazzaglia et al., 2022) or state entropy (Lee et al., 2019; Pong et al., 2020; Liu & Abbeel, 2021b; Yarats et al., 2021). Based on the data collected by the exploration

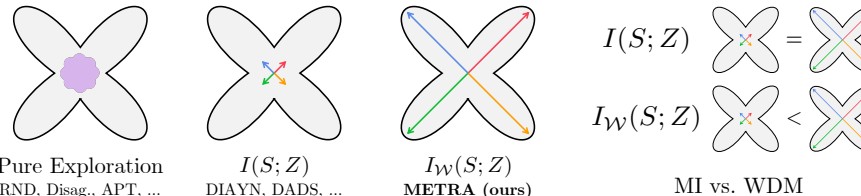

Figure 2: **Sketch comparing different unsupervised RL objectives.** Pure exploration approaches try to cover every possible state, which is infeasible in complex environments (*e.g.*, such methods might be "stuck" at forever finding novel joint angle configurations of a robot, without fully exploring the environment; see Figure 3). The mutual information $I(S;Z)$ has no underlying distance metrics, and thus does not prioritize coverage enough, only focusing on skills that are discriminable. In contrast, our proposed Wasserstein dependency measure $I_{\mathcal{W}}(S;Z)$ maximizes the distance metric $d$, which we choose to be the temporal distance, forcing the learned skills to span the "longest" subspaces of the state space, analogously to (temporal, nonlinear) PCA.

policy, these methods learn a world model (Rajeswar et al., 2023), train a goal-conditioned policy (Pong et al., 2020; Pitis et al., 2020; Mendonca et al., 2021; Hu et al., 2023), learn skills via trajectory autoencoders (Campos Camúñez et al., 2020; Mazzaglia et al., 2023), or directly fine-tune the learned exploration policy (Laskin et al., 2021) to accelerate downstream task learning. While these pure exploration-based approaches are currently the leading methods in unsupervised RL benchmarks (Mendonca et al., 2021; Laskin et al., 2021; Mazzaglia et al., 2023; Rajeswar et al., 2023), their scalability may be limited in complex environments with large state spaces because it is often computationally infeasible to completely cover every possible state or fully capture the dynamics. In Section 5, we empirically demonstrate that these approaches even fail to cover the state space of the state-based 29-dimensional Ant environment.

Another line of research in unsupervised RL aims to learn diverse behaviors (or *skills*) that are distinguishable from one another, and our method also falls into this category. The most common approach to unsupervised skill discovery is to maximize the mutual information (MI) between states and skills (Gregor et al., 2016; Eysenbach et al., 2019a; Sharma et al., 2020; Hansen et al., 2020):

$$I(S;Z) = D_{\mathrm{KL}}(p(s,z)\|p(s)p(z)). \tag{1}$$

By associating different skill latent vectors $z$ with different states $s$, these methods learn diverse skills that are mutually distinct. However, they share the limitation that they often end up discovering simple, static behaviors with limited state coverage (Campos Camúñez et al., 2020; Park et al., 2022). This is because MI is defined by a KL divergence (Equation (1)), which is a *metric-agnostic* quantity (*e.g.*, MI is invariant to scaling; see Figure 2). As a result, the MI objective only focuses on the distinguishability of behaviors, regardless of "how different" they are, resulting in limited state coverage (Campos Camúñez et al., 2020; Park et al., 2022). To address this limitation, prior works combine the MI objective with exploration bonuses (Campos Camúñez et al., 2020; Strouse et al., 2022; Park & Levine, 2023) or propose different objectives that encourage maximizing distances in the state space (He et al., 2022; Park et al., 2022; 2023b). Yet, it remains unclear whether these methods can scale to complex, high-dimensional environments, because they either attempt to completely capture the entire MDP (Campos Camúñez et al., 2020; Strouse et al., 2022; Park & Levine, 2023) or assume a compact, structured state space (He et al., 2022; Park et al., 2022; 2023b). Indeed, to the best of our knowledge, *no* previous unsupervised skill discovery methods have succeeded in discovering locomotion behaviors on pixel-based locomotion environments. Unlike these approaches, our method learns a compact set of diverse behaviors that are maximally different in terms of the *temporal distance*. As a result, they can approximately cover the state space, even in a complex, high-dimensional environment. We discuss further related work in Appendix B.

## 3 PRELIMINARIES AND PROBLEM SETTING

We consider a controlled Markov process, an MDP without a reward function, defined as $\mathcal{M} = (\mathcal{S}, \mathcal{A}, \mu, p)$. $\mathcal{S}$ denotes the state space, $\mathcal{A}$ denotes the action space, $\mu : \Delta(\mathcal{S})$ denotes the initial state distribution, and $p : \mathcal{S} \times \mathcal{A} \to \Delta(\mathcal{A})$ denotes the transition dynamics kernel. We consider a set of latent vectors $z \in \mathcal{Z}$, which can be either discrete or continuous, and a latent-conditioned policy $\pi(a|s,z)$. Following the terminology in unsupervised skill discovery, we refer to latent vectors $z$ (and their corresponding policies $\pi(a|s,z)$) as *skills*. When sampling a trajectory, we first sample a skill from the prior distribution, $z \sim p(z)$, and then roll out a trajectory with $\pi(a|s,z)$, where $z$ is fixed for the entire episode. Hence, the joint skill-trajectory distribution is given as $p(\tau, z) =$

$p(z)p(s_0)\prod_{t=0}^{T-1}\pi(a_t|s_t,z)p(s_{t+1}|s_t,a_t)$, where $\tau$ denotes $(s_0, a_0, s_1, a_1, \ldots, s_T)$. Our goal in this work is to learn a set of diverse, useful behaviors $\pi(a|s,z)$, without using any supervision, data, or prior knowledge.

## 4    A Scalable Objective for Unsupervised RL

**Desiderata.** We first state our two desiderata for a scalable unsupervised RL objective. First, instead of covering every possible state in a given MDP, which is infeasible in complex environments, we want to have a compact latent space $\mathcal{Z}$ of a *tractable* size and a latent-conditioned policy $\pi(a|s,z)$ that translates latent vectors into actual behaviors. Second, we want the behaviors from different latent vectors to be different, collectively covering as much of the state space as possible. In other words, we want to **maximize state coverage under the given capacity** of $\mathcal{Z}$. An algorithm that satisfies these two desiderata would be scalable to complex environments, because we only need to learn a compact set of behaviors that approximately cover the MDP.

**Objective.** Based on the above, we propose the following novel objective for unsupervised RL:

$$I_{\mathcal{W}}(S;Z) = \mathcal{W}(p(s,z), p(s)p(z)), \tag{2}$$

where $I_{\mathcal{W}}(S;Z)$ is the Wasserstein dependency measure (WDM) (Ozair et al., 2019) between states and skills, and $\mathcal{W}$ is the 1-Wasserstein distance on the metric space $(\mathcal{S} \times \mathcal{Z}, d)$ with a *distance metric* $d$. Intuitively, the WDM objective in Equation (2) can be viewed as a "Wasserstein variant" of the previous MI objective (Equation (1)), where the KL divergence in MI is replaced with the Wasserstein distance. However, despite the apparent similarity, there exists a significant difference between the two objectives: MI is completely agnostic to the underlying distance metric, while WDM is a *metric-aware* quantity. As a result, the WDM objective (Equation (2)) not only discovers diverse skills that are different from one another, as in the MI objective, but also actively maximizes distances $d$ between different skill trajectories (Figure 2). This makes them collectively cover the state space as much as possible (in terms of the given metric $d$). The choice of metric for $d$ is critical for effective skill discovery, and simple choices like Euclidean metrics on the state space would generally *not* be effective for non-metric state representations, such as images. Therefore, instantiating this approach with the right metric is an important part of our contribution, as we will discuss in Section 4.2. Until then, we assume that we have a given metric $d$.

### 4.1    Tractable Optimization

While our objective $I_{\mathcal{W}}(S;Z)$ has several desirable properties, it is not immediately straightforward to maximize this quantity in practice. In this section, we describe a simple, tractable objective that can be used to maximize $I_{\mathcal{W}}(S;Z)$ in practice. We begin with the Kantorovich-Rubenstein duality (Villani et al., 2009; Ozair et al., 2019), which provides a tractable way to maximize the Wasserstein dependency measure:

$$I_{\mathcal{W}}(S;Z) = \sup_{\|f\|_L \leq 1} \mathbb{E}_{p(s,z)}[f(s,z)] - \mathbb{E}_{p(s)p(z)}[f(s,z)], \tag{3}$$

where $\|f\|_L$ denotes the Lipschitz constant for the function $f : \mathcal{S} \times \mathcal{Z} \to \mathbb{R}$ under the given distance metric $d$, *i.e.*, $\|f\|_L = \sup_{(s_1,z_1) \neq (s_2,z_2)} |f(s_1,z_1) - f(s_2,z_2)|/d((s_1,z_1),(s_2,z_2))$. Intuitively, $f$ is a score function that assigns larger values to $(s, z)$ tuples sampled from the joint distribution and smaller values to $(s, z)$ tuples sampled independently from their marginal distributions. We note that Equation (3) is already a tractable objective, as we can jointly train a 1-Lipschitz-constrained score function $f(s, z)$ using gradient descent and a skill policy $\pi(a|s,z)$ using RL, with the reward function being an empirical estimate of Equation (3), $r(s,z) = f(s,z) - N^{-1}\sum_{i=1}^{N} f(s,z_i)$, where $z_1, z_2, \ldots, z_N$ are $N$ independent random samples from the prior distribution $p(z)$.

However, since sampling $N$ additional $z$s for each data point is computationally demanding, we will further simplify the objective to enable more efficient learning. First, we consider the parameterization $f(s,z) = \phi(s)^\top \psi(z)$ with $\phi : \mathcal{S} \to \mathbb{R}^D$ and $\psi : \mathcal{Z} \to \mathbb{R}^D$ with independent 1-Lipschitz constraints[1], which yields the following objective:

$$I_{\mathcal{W}}(S;Z) \approx \sup_{\|\phi\|_L \leq 1, \|\psi\|_L \leq 1} \mathbb{E}_{p(s,z)}[\phi(s)^\top \psi(z)] - \mathbb{E}_{p(s)}[\phi(s)]^\top \mathbb{E}_{p(z)}[\psi(z)]. \tag{4}$$

---

[1]While $\|\phi\|_L \leq 1, \|\psi\|_L \leq 1$ is not technically equivalent to $\|f\|_L \leq 1$, we use the former as it is more tractable. Also, we note that $\|f\|_L$ can be upper-bounded in terms of $\|\phi\|_L$, $\|\psi\|_L$, $\sup_s \|\phi(s)\|_2$, and $\sup_z \|\psi(z)\|_2$ under $d((s_1,z_1),(s_2,z_2)) = (\sup_s \|\phi(s)\|_2)\|\psi\|_L d(z_1,z_2) + (\sup_z \|\psi(z)\|_2)\|\phi\|_L d(s_1,s_2)$.

Here, we note that the decomposition $f(s, z) = \phi(s)^\top \psi(z)$ is *universal*; *i.e.*, the expressiveness of $f(s, z)$ is equivalent to that of $\phi(s)^\top \psi(z)$ when $D \to \infty$. The proof can be found in Appendix C.

Next, we consider a variant of the Wasserstein dependency measure that only depends on the last state: $I_\mathcal{W}(S_T; Z)$, similarly to VIC (Gregor et al., 2016). This allows us to further decompose the objective with a telescoping sum as follows:

$$I_\mathcal{W}(S_T; Z) \approx \sup_{\|\phi\|_L \leq 1, \|\psi\|_L \leq 1} \mathbb{E}_{p(\tau,z)}[\phi(s_T)^\top \psi(z)] - \mathbb{E}_{p(\tau)}[\phi(s_T)]^\top \mathbb{E}_{p(z)}[\psi(z)] \tag{5}$$

$$= \sup_{\phi,\psi} \sum_{t=0}^{T-1} \left( \mathbb{E}_{p(\tau,z)}[(\phi(s_{t+1}) - \phi(s_t))^\top \psi(z)] - \mathbb{E}_{p(\tau)}[\phi(s_{t+1}) - \phi(s_t)]^\top \mathbb{E}_{p(z)}[\psi(z)] \right), \tag{6}$$

where we also use the fact that $p(s_0)$ and $p(z)$ are independent. Finally, we set $\psi(z)$ to $z$. While this makes $\psi$ less expressive, it allows us to derive the following concise objective:

$$I_\mathcal{W}(S_T; Z) \approx \sup_{\|\phi\|_L \leq 1} \mathbb{E}_{p(\tau,z)} \left[ \sum_{t=0}^{T-1} (\phi(s_{t+1}) - \phi(s_t))^\top (z - \bar{z}) \right], \tag{7}$$

where $\bar{z} = \mathbb{E}_{p(z)}[z]$. Here, since we can always shift the prior distribution $p(z)$ to have a zero mean, we can assume $\bar{z} = 0$ without loss of generality. This objective can now be easily maximized by jointly training $\phi(s)$ and $\pi(a|s, z)$ with $r(s, z, s') = (\phi(s') - \phi(s))^\top z$ under the constraint $\|\phi\|_L \leq 1$. Note that we do not need any additional random samples of $z$, unlike Equation (3).

## 4.2 FULL OBJECTIVE: METRIC-AWARE ABSTRACTION (METRA)

So far, we have not specified the distance metric $d$ for the Wasserstein distance in WDM (or equivalently for the Lipschitz constraint $\|\phi\|_L \leq 1$). Choosing an appropriate distance metric is crucial for learning a compact set of useful behaviors, because it determines the *priority* by which the behaviors are learned within the capacity of $\mathcal{Z}$. Previous metric-based skill discovery methods mostly employed the Euclidean distance (or its scaled variant) as a metric (He et al., 2022; Park et al., 2022; 2023b). However, they are not directly scalable to high-dimensional environments with pixel-based observations, in which the Euclidean distance is not necessarily meaningful.

In this work, we propose to use the *temporal distance* (Kaelbling, 1993; Hartikainen et al., 2020; Durugkar et al., 2021) between two states as a distance metric $d_{\text{temp}}(s_1, s_2)$, the minimum number of environment steps to reach $s_2$ from $s_1$. This provides a natural way to measure the distance between two states, as it only depends on the *inherent* transition dynamics of the MDP, being invariant to the state representation and thus scalable to pixel-based environments. Using the temporal distance metric, we can rewrite Equation (7) as follows:

$$\sup_{\pi,\phi} \mathbb{E}_{p(\tau,z)} \left[ \sum_{t=0}^{T-1} (\phi(s_{t+1}) - \phi(s_t))^\top z \right] \quad \text{s.t.} \quad \|\phi(s) - \phi(s')\|_2 \leq 1, \quad \forall (s, s') \in \mathcal{S}_{\text{adj}}, \tag{8}$$

where $\mathcal{S}_{\text{adj}}$ denotes the set of adjacent state pairs in the MDP. Note that $\|\phi\|_L \leq 1$ is equivalently converted into $\|\phi(s) - \phi(s')\|_2 \leq 1$ under the temporal distance metric (see Theorem C.3).

**Intuition and interpretation.** We next describe how the constrained objective in Equation (8) may be interpreted. Intuitively, a policy $\pi(a|s, z)$ that maximizes our objective should learn to move as far as possible along various directions in the latent space (specified by $z$). Since distances in the latent space, $\|\phi(s_1) - \phi(s_2)\|_2$, are always upper-bounded by the corresponding temporal distances in the MDP, given by $d_{\text{temp}}(s_1, s_2)$, the learned latent space should assign its (limited) dimensions to the manifolds in the original state space that are maximally "spread out", in the sense that shortest paths within the set of represented states should be as long as possible. This conceptually resembles "principal components" of the state space, but with respect to shortest paths rather than Euclidean distances, and with non-linear $\phi$ rather than linear $\phi$. Thus, we would expect $\phi$ to learn to abstract the state space in a lossy manner, preserving temporal distances (Figure 9), and emphasizing those degrees of freedom of the state that span the largest possible "temporal" (non-linear) manifolds (Figure 1). Based on this intuition, we call our method **Metric-Aware Abstraction** (**METRA**). In Appendix D, we derive a formal connection between METRA and principal component analysis (PCA) under the temporal distance metric under several simplifying assumptions.

**Theorem 4.1** (Informal statement of Theorem D.2). *Under some simplifying assumptions, linear squared METRA is equivalent to PCA under the temporal distance metric.*

---

**Algorithm 1** Metric-Aware Abstraction (METRA)

---

1: Initialize skill policy $\pi(a|s, z)$, representation function $\phi(s)$, Lagrange multiplier $\lambda$, replay buffer $\mathcal{D}$
2: **for** $i \leftarrow 1$ to (# epochs) **do**
3:     **for** $j \leftarrow 1$ to (# episodes per epoch) **do**
4:         Sample skill $z \sim p(z)$
5:         Sample trajectory $\tau$ with $\pi(a|s, z)$ and add to replay buffer $\mathcal{D}$
6:     **end for**
7:     Update $\phi(s)$ to maximize $\mathbb{E}_{(s,z,s')\sim\mathcal{D}}[(\phi(s') - \phi(s))^\top z + \lambda \cdot \min(\varepsilon, 1 - \|\phi(s) - \phi(s')\|_2^2)]$
8:     Update $\lambda$ to minimize $\mathbb{E}_{(s,z,s')\sim\mathcal{D}}[\lambda \cdot \min(\varepsilon, 1 - \|\phi(s) - \phi(s')\|_2^2)]$
9:     Update $\pi(a|s, z)$ using SAC (Haarnoja et al., 2018a) with reward $r(s, z, s') = (\phi(s') - \phi(s))^\top z$
10: **end for**

---

**Connections to previous skill discovery methods.** There exist several intriguing connections between our WDM objective (Equation (2)) and previous skill discovery methods, including DI-AYN (Eysenbach et al., 2019a), DADS (Sharma et al., 2020), CIC (Laskin et al., 2022), LSD (Park et al., 2022), and CSD (Park et al., 2023b). Perhaps the most apparent connections are with LSD and CSD, which also use similar constrained objectives to Equation (7). In fact, although not shown by the original authors, the constrained inner product objectives of LSD and CSD are also equivalent to $I_\mathcal{W}(S_T; Z)$, but with the Euclidean distance (or its normalized variant), instead of the temporal distance. Also, the connection between WDM and Equation (7) provides further theoretical insight into the rather "ad-hoc" choice of zero-centered one-hot vectors used in discrete LSD (Park et al., 2022); we must use a zero-mean prior distribution due to the $z - \bar{z}$ term in Equation (7). There exist several connections between our WDM objective and previous MI-based skill discovery methods as well. Specifically, by simplifying WDM (Equation (2)) in three different ways, we can obtain "Wasserstein variants" of DIAYN, DADS, and CIC. We refer to Appendix E for detailed derivations.

**Zero-shot goal-reaching with METRA.** Thanks to the state abstraction function $\phi(s)$, METRA provides a simple way to command the skill policy to reach a goal state in a *zero-shot* manner, as in LSD (Park et al., 2022). Since $\phi$ abstracts the state space preserving temporal distances, the difference vector $\phi(g) - \phi(s)$ tells us the skill we need to select to reach the goal state $g$ from the current state $s$. As such, by simply setting $z = (\phi(g) - \phi(s))/\|\phi(g) - \phi(s)\|_2$ (for continuous skills) or $z = \arg\max_{\dim} (\phi(g) - \phi(s))$ (for discrete skills), we can find the skill that leads to the goal. With this technique, METRA can solve goal-conditioned tasks without learning a separate goal-conditioned policy, as we will show in Section 5.3.

**Implementation.** We optimize the constrained objective in Equation (8) using dual gradient descent with a Lagrange multiplier $\lambda$ and a small relaxation constant $\varepsilon > 0$, similarly to Park et al. (2023b); Wang et al. (2023). We provide a pseudocode for METRA in Algorithm 1.

**Limitations.** One potential issue with Equation (8) is that we embed the temporal distance into the symmetric Euclidean distance in the latent space, where the temporal distance can be asymmetric. This makes our temporal distance abstraction more "conservative" in the sense that it considers the minimum of both temporal distances, *i.e.*, $\|\phi(s_1) - \phi(s_2)\|_2 \leq \min(d_{\text{temp}}(s_1, s_2), d_{\text{temp}}(s_2, s_1))$. While this conservatism is less problematic in our benchmark environments, in which transitions are mostly "symmetric", it might be overly restrictive in highly asymmetric environments. To resolve this, we can replace the Euclidean distance $\|\phi(s_1) - \phi(s_2)\|_2$ in Equation (8) with an asymmetric *quasimetric*, as in Wang et al. (2023). We leave this extension for future work. Another limitation is that the simplified WDM objective in Equation (7) only considers behaviors that move linearly in the latent space. While this does not necessarily imply that the behaviors are also linear in the original state space (because $\phi : \mathcal{S} \to \mathcal{Z}$ is a nonlinear mapping), this simplification, which stems from the fact that we set $\psi(z) = z$, might restrict the diversity of behaviors to some degree. We believe this can be addressed by using the full WDM objective in Equation (4). Notably, the full objective (Equation (4)) resembles contrastive learning, and we believe combining it with scalable contrastive learning techniques is an exciting future research direction (see Appendix E.3). We refer to Appendix A for practical limitations regarding our implementation of METRA.

## 5 EXPERIMENTS

Through our experiments in benchmark environments, we aim to answer the following questions: (1) Can METRA scale to complex, high-dimensional environments, including domains with image observations? (2) Does METRA discover meaningful behaviors in complex environments with no supervision? (3) Are the behaviors discovered by METRA useful for downstream tasks?

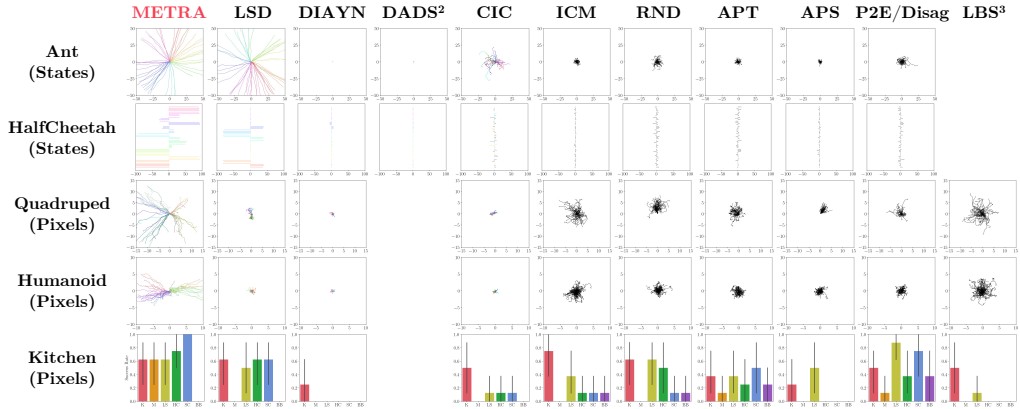

Figure 3: **Examples of behaviors learned by 11 unsupervised RL methods.** For locomotion environments, we plot the $x$-$y$ (or $x$) trajectories sampled from learned policies. For Kitchen, we measure the coincidental success rates for six predefined tasks. Different colors represent different skills $z$. METRA is the *only* method that discovers diverse locomotion skills in pixel-based Quadruped and Humanoid. We refer to Figure 11 for the complete qualitative results (8 seeds) of METRA and our project page for videos.

## 5.1 EXPERIMENTAL SETUP

We evaluate our method on five robotic locomotion and manipulation environments (Figure 4): state-based Ant and HalfCheetah from Gym (Todorov et al., 2012; Brockman et al., 2016), pixel-based Quadruped and Humanoid from the DeepMind Control (DMC) Suite (Tassa et al., 2018), and a pixel-based version of Kitchen

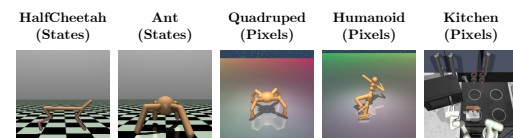

Figure 4: **Benchmark environments.**

from Gupta et al. (2019); Mendonca et al. (2021). For pixel-based DMC locomotion environments, we use colored floors to allow the agent to infer its location from pixels, similarly to Hafner et al. (2022); Park et al. (2023a) (Figure 10). Throughout the experiments, we do *not* use any prior knowledge, data, or supervision (*e.g.*, observation restriction, early termination, etc.). As such, in pixel-based environments, the agent must learn diverse behaviors solely from $64 \times 64 \times 3$ camera images.

We compare METRA against 11 previous methods in three groups: (1) unsupervised skill discovery, (2) unsupervised exploration, and (3) unsupervised goal-reaching methods. For unsupervised skill discovery methods, we compare against two MI-based approaches, DIAYN (Eysenbach et al., 2019a) and DADS (Sharma et al., 2020), one hybrid method that combines MI and an exploration bonus, CIC (Laskin et al., 2022), and one metric-based approach that maximizes Euclidean distances, LSD (Park et al., 2022). For unsupervised exploration methods, we consider five pure exploration approaches, ICM (Pathak et al., 2017), RND (Burda et al., 2019), Plan2Explore (Sekar et al., 2020) (or Disagreement (Pathak et al., 2019)), APT (Liu & Abbeel, 2021b), and LBS (Mazzaglia et al., 2022), and one hybrid approach that combines exploration and successor features, APS (Liu & Abbeel, 2021a). We note that the Dreamer (Hafner et al., 2020) variants of these methods (especially LBS (Mazzaglia et al., 2022)) are currently the state-of-the-art methods in the pixel-based unsupervised RL benchmark (Laskin et al., 2021; Rajeswar et al., 2023). For unsupervised goal-reaching methods, we mainly compare with a state-of-the-art unsupervised RL approach, LEXA (Mendonca et al., 2021), as well as two previous skill discovery methods that enable zero-shot goal-reaching, DIAYN and LSD. We use 2-D skills for Ant and Humanoid, 4-D skills for Quadruped, 16 discrete skills for HalfCheetah, and 24 discrete skills for Kitchen. For CIC, we use 64-D skill latent vectors for all environments, following the original suggestion (Laskin et al., 2022).

## 5.2 QUALITATIVE COMPARISON

We first demonstrate examples of behaviors (or skills) learned by our method and the 10 prior unsupervised RL methods on each of the five benchmark environments in Figure 3. The figure illustrates that METRA discovers diverse behaviors in both state-based and pixel-based domains. Notably, METRA is the only method that successfully discovers locomotion skills in *pixel-based* Quadruped and Humanoid, and shows qualitatively very different behaviors from previous unsupervised RL methods across the environments. Pure exploration methods mostly exhibit chaotic, random behaviors (videos), and fail to fully explore the state space (in terms of $x$-$y$ coordinates) even in state-based

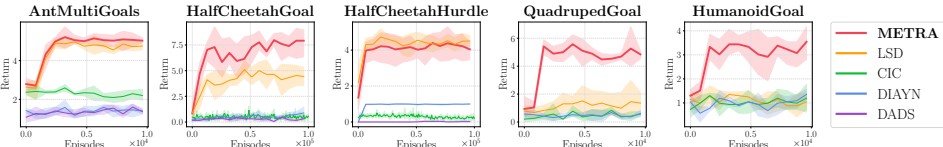

Figure 5: **Quantitative comparison with unsupervised skill discovery methods (8 seeds).** We measure the state/task coverage of the policies learned by five skill discovery methods. METRA exhibits the best coverage across all environments, while previous methods completely fail to explore the state spaces of pixel-based locomotion environments. Notably, METRA is the *only* method that discovers locomotion skills in pixel-based Quadruped and Humanoid.

Figure 6: **Downstream task performance comparison of unsupervised skill discovery methods (4 seeds).** To verify whether learned skills are useful for downstream tasks, we train a hierarchical high-level controller on top of the frozen skill policy to maximize task rewards. METRA exhibits the best or near-best performance across the five tasks, which suggests that the behaviors learned by METRA are indeed useful for the tasks.

Ant and HalfCheetah. This is because it is practically infeasible to completely cover the infinitely many combinations of joint angles and positions in these domains. MI-based skill discovery methods also fail to explore large portions of the state space due to the metric-agnosticity of the KL divergence (Section 2), even when combined with an exploration bonus (*i.e.*, CIC). LSD, a previous metric-based skill discovery method that maximizes Euclidean distances, does discover locomotion skills in state-based environments, but fails to scale to the pixel-based environments, where the Euclidean distance on image pixels does not necessarily provide a meaningful metric. In contrast to these methods, METRA learns various task-related behaviors by maximizing temporal distances in diverse ways. On our project page, we show additional qualitative results of METRA with different skill spaces. We note that, when combined with a discrete latent space, METRA discovers even more diverse behaviors, such as doing a backflip and taking a static posture, in addition to locomotion skills. We refer to Appendix F for visualization of learned latent spaces of METRA.

## 5.3 QUANTITATIVE COMPARISON

Next, we quantitatively compare METRA against three groups of 11 previous unsupervised RL approaches, using different metrics that are tailored to each group's primary focus. For quantitative results, we use 8 seeds and report $95\%$ confidence intervals, unless otherwise stated.

**Comparison with unsupervised skill discovery methods.** We first compare METRA with other methods that also aim to solve the skill discovery problem (i.e., learning a latent-conditioned policy $\pi(a|s,z)$ that performs different skills for different values of $z$). These include LSD, CIC, DIAYN, and DADS[2]. We implement these methods on the same codebase as METRA. For comparison, we employ two metrics: policy coverage and downstream task performance. Figure 5 presents the policy coverage results, where we evaluate the skill policy's $x$ coverage (HalfCheetah), $x$-$y$ coverage (Ant, Quadruped, and Humanoid), or task (Kitchen) coverage at each evaluation epoch. The results show that METRA achieves the best performance in most of the domains, and is the only method that successfully learns meaningful skills in the pixel-based settings, where previous skill discovery methods generally fail. In Figure 6, we evaluate the applicability of the skills discovered by each method to downstream tasks, where the downstream task is learned by a hierarchical controller $\pi^h(z|s)$ that selects (frozen) learned skills to maximize the task reward (see Appendix G for details). METRA again achieves the best performance on most of these tasks, suggesting that the behaviors learned by METRA not only provide greater coverage, but also are more suitable for downstream tasks in these domains.

**Comparison with pure exploration methods.** Next, we quantitatively compare METRA to five unsupervised exploration methods, which do not aim to learn skills but only attempt to cover the state space, ICM, LBS[3], RND, APT, and Plan2Explore (or Disagreement), and one hybrid method

---

[2]We do not compare against DADS in pixel-based environments due to the computational cost of its skill dynamics model $p(s'|s,z)$, which requires predicting the full next image.

[3]Since LBS requires a world model, we only evaluate it on pixel-based environments, where we use the Dreamer variants of pure exploration methods (Rajeswar et al., 2023).

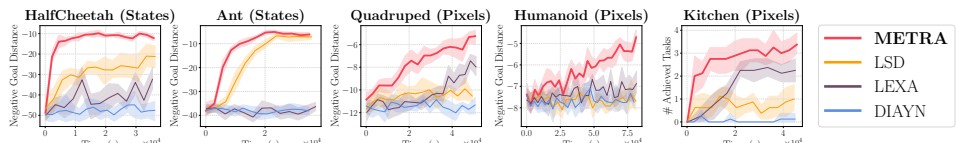

Figure 7: **Quantitative comparison with pure exploration methods (8 seeds).** We compare METRA with six unsupervised exploration methods in terms of state coverage. Since it is practically infeasible to completely cover every possible state or transition, pure exploration methods struggle to explore the state space of complex environments, such as pixel-based Humanoid or state-based Ant.

Figure 8: **Downstream task performance comparison with LEXA (8 seeds).** We compare METRA against LEXA, a state-of-the-art unsupervised goal-reaching method, on five goal-conditioned tasks. The skills learned by METRA can be employed to solve these tasks in a zero-shot manner, achieving the best performance.

that combines exploration and successor features, APS. We use the original implementations by Laskin et al. (2021) for state-based environments and the Dreamer versions by Rajeswar et al. (2023) for pixel-based environments. As the underlying RL backbones are very different (*e.g.*, Dreamer is model-based, while METRA uses model-free SAC), we compare the methods based on wall clock time. For the metric, instead of policy coverage (as in Figure 5), we measure *total* state coverage (*i.e.*, the number of bins covered by any *training* trajectories up to each evaluation epoch). This metric is more generous toward the exploration methods, since such methods might not cover the entire space on any single iteration, but rather visit different parts of the space on different iterations (in contrast to our method, which aims to produce diverse skills). In Kitchen, we found that most methods max out the total task coverage metric, and we instead use both the queue coverage and policy coverage metrics (see Appendix G for details). Figure 7 presents the results, showing that METRA achieves the best coverage in most of the environments. While pure exploration methods also work decently in the pixel-based Kitchen, they fail to fully explore the state spaces of state-based Ant and pixel-based Humanoid, which have complex dynamics with nearly infinite possible combinations of positions, joint angles, and velocities.

**Comparison with unsupervised goal-reaching methods.** Finally, we compare METRA with LEXA, a state-of-the-art unsupervised goal-reaching method. LEXA trains an exploration policy with Plan2Explore (Sekar et al., 2020), which maximizes epistemic uncertainty in the transition dynamics model, in parallel with a goal-conditioned policy $\pi(a|s, g)$ on the data collected by the exploration policy. We compare the performances of METRA, LEXA, and two previous skill discovery methods (DIAYN and LSD) on five goal-reaching downstream tasks. We use the procedure described in Section 4.2 to solve goal-conditioned tasks in a zero-shot manner with METRA. Figure 8 presents the comparison results, where METRA achieves the best performance on all of the five downstream tasks. While LEXA also achieves non-trivial performances in three tasks, it struggles with state-based Ant and pixel-based Humanoid, likely because it is practically challenging to completely capture the transition dynamics of these complex environments.

## 6 CONCLUSION

In this work, we presented METRA, a scalable unsupervised RL method based on the idea of covering a compact latent skill space that is connected to the state space by a temporal distance metric. We showed that METRA learns diverse useful behaviors in various locomotion and manipulation environments, being the first unsupervised RL method that learns locomotion behaviors in pixel-based Quadruped and Humanoid.

**Final remarks.** In unsupervised RL, many excellent prior works have explored pure exploration or mutual information skill learning. However, given that these methods are not necessarily readily scalable to complex environments with high intrinsic state dimensionality, as discussed in Section 2, we may need a completely different approach to enable truly scalable unsupervised RL. We hope that this work serves as a step toward broadly applicable unsupervised RL that enables large-scale pre-training with minimal supervision.

## ACKNOWLEDGMENTS

We would like to thank Youngwoon Lee for an informative discussion, and RAIL members and anonymous reviewers for their helpful comments. This work was partly supported by the Korea Foundation for Advanced Studies (KFAS), ARO W911NF-21-1-0097, and the Office of Naval Research. This research used the Savio computational cluster resource provided by the Berkeley Research Computing program at UC Berkeley.

## REPRODUCIBILITY STATEMENT

We provide our code at the following repository: https://github.com/seohongpark/METRA. We provide the full experimental details in Appendix G.

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

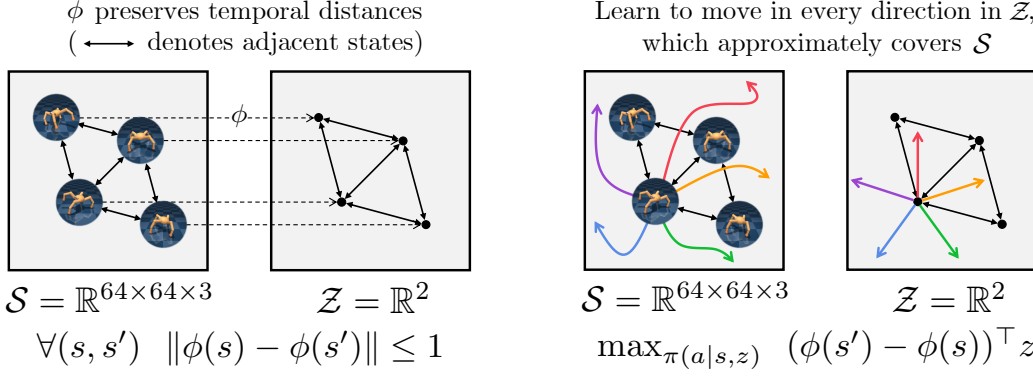

**Figure 9: Intuitive interpretation of METRA.** Our main idea is to only cover a compact latent space $\mathcal{Z}$ that is *metrically* connected to the state space $\mathcal{S}$. Specifically, METRA learns a lossy, compact representation function $\phi : \mathcal{S} \to \mathcal{Z}$, which preserves *temporal* distances (*left*), and learns to move in every direction in $\mathcal{Z}$, which leads to approximate coverage of the state space $\mathcal{S}$ (*right*).

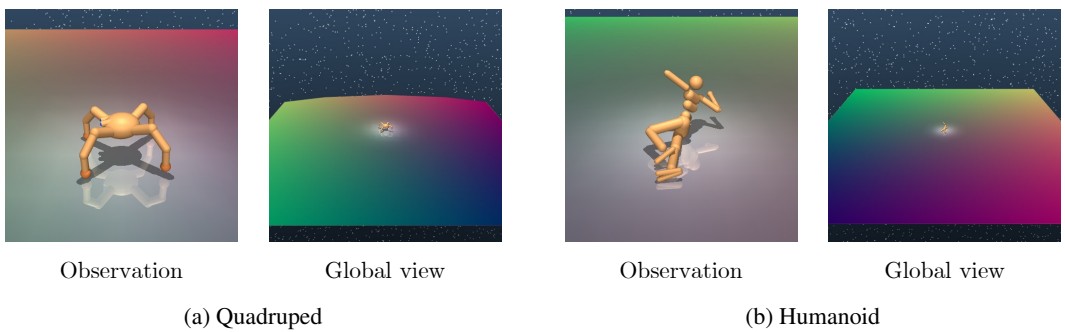

|  |  |
|:--:|:--:|
| Observation | Global view |
| Observation | Global view |
| (a) Quadruped | (b) Humanoid |

**Figure 10: Visualization of pixel-based DMC Quadruped and Humanoid.** We use gradient-colored floors to allow the agent to infer its location from pixel observations, similarly to Hafner et al. (2022); Park et al. (2023a).

## A  LIMITATIONS

Despite its state-of-the-art performance in several benchmark environments, METRA, in its current form, has limitations. We refer to Section 4.2 for the limitations and future research directions regarding the METRA objective. In terms of practical implementation, METRA, like other similar unsupervised skill discovery methods (Sharma et al., 2020; Park et al., 2022; 2023b), uses a relatively small update-to-data (UTD) ratio (*i.e.*, the average number of gradient steps per environment step); *e.g.*, we use $1/4$ for Kitchen and $1/16$ for Quadruped and Humanoid. Although we demonstrate that METRA learns efficiently in terms of wall clock time, we believe there is room for improvement in terms of sample efficiency. This is mainly because we use vanilla SAC (Haarnoja et al., 2018a) as its RL backbone for simplicity, and we believe increasing the sample efficiency of METRA by combining it with recent techniques in model-free RL (Kostrikov et al., 2021; Chen et al., 2021; Hiraoka et al., 2022) or model-based RL (Hafner et al., 2020; Hansen et al., 2022) is an interesting direction for future work.

Another limitation of this work is that, while we evaluate METRA on various locomotion and manipulation environments, following prior work in unsupervised RL and unsuperivsed skill discovery (Eysenbach et al., 2019a; Sharma et al., 2020; Mendonca et al., 2021; Laskin et al., 2021; He et al., 2022; Park et al., 2022; Zhao et al., 2022; Shafiullah & Pinto, 2022; Laskin et al., 2022; Park et al., 2023b; Yang et al., 2023), we have not evaluated METRA on other different types of environments, such as Atari games. Also, since we assume a fixed MDP (*i.e.*, stationary, fully observable dynamics, Section 3), METRA in its current form does not particularly deal with non-stationary or

non-Markovian dynamics. We leave applying METRA to more diverse environments or extending the idea behind METRA to non-stationary or non-Markovian environments for future work.

# B  EXTENDED RELATED WORK

In addition to unsupervised skill discovery (Mohamed & Rezende, 2015; Gregor et al., 2016; Florensa et al., 2017; Co-Reyes et al., 2018; Achiam et al., 2018; Eysenbach et al., 2019a; Warde-Farley et al., 2019; Shyam et al., 2019; Lee et al., 2019; Sharma et al., 2020; Campos Camúñez et al., 2020; Hansen et al., 2020; Pong et al., 2020; Baumli et al., 2021; Choi et al., 2021; Yarats et al., 2021; Kim et al., 2021; Zhang et al., 2021; He et al., 2022; Strouse et al., 2022; Laskin et al., 2022; Park et al., 2022; Shafiullah & Pinto, 2022; Jiang et al., 2022; Zhao et al., 2022; Kamienny et al., 2022; Park & Levine, 2023; Park et al., 2023b; Li et al., 2023; Kim et al., 2023) and pure exploration (or unsupervised goal-conditioned RL) methods (Houthooft et al., 2016; Bellemare et al., 2016; Tang et al., 2017; Ostrovski et al., 2017; Fu et al., 2017; Pathak et al., 2017; Hazan et al., 2019; Shyam et al., 2019; Burda et al., 2019; Pathak et al., 2019; Lee et al., 2019; Ecoffet et al., 2020; Pitis et al., 2020; Badia et al., 2020; Mutti et al., 2021; Liu & Abbeel, 2021b; Mendonca et al., 2021; Yarats et al., 2021; Seo et al., 2021; Mazzaglia et al., 2022; 2023; Hu et al., 2023; Rajeswar et al., 2023), there have also been proposed other types of unsupervised RL approaches, such as ones based on asymmetric self-play (Sukhbaatar et al., 2018; OpenAI et al., 2021), surprise minimization (Berseth et al., 2021; Rhinehart et al., 2021), and forward-backward representations (Touati & Ollivier, 2021; Touati et al., 2023). One potentially closely related line of work is graph Laplacian-based option discovery methods (Machado et al., 2017; 2018; Klissarov & Machado, 2023). These methods learn a set of diverse behaviors based on the eigenvectors of the graph Laplacian of the MDP's adjacency matrix. Although we have not found a formal connection to these methods, we suspect there might exist a deep, intriguing connection between METRA and graph Laplacian-based methods, given that they both discover behaviors based on the temporal dynamics of the MDP. METRA is also related to several works in goal-conditioned RL that consider temporal distances (Kaelbling, 1993; Schaul et al., 2015; Savinov et al., 2018; Eysenbach et al., 2019b; Florensa et al., 2019; Hartikainen et al., 2020; Durugkar et al., 2021; Wang et al., 2023). In particular, Durugkar et al. (2021); Wang et al. (2023) use similar temporal distance constraints to ours for goal-conditioned RL.

# C  THEORETICAL RESULTS

## C.1  UNIVERSALITY OF INNER PRODUCT DECOMPOSITION

**Lemma C.1.** *Let $\mathcal{X}$ and $\mathcal{Y}$ be compact Hausdorff spaces (e.g., compact subsets in $\mathbb{R}^N$) and $\mathcal{C}(\mathcal{A})$ be the set of real-valued continuous functions on $\mathcal{A}$. For any function $f(x, y) \in \mathcal{C}(\mathcal{X} \times \mathcal{Y})$ and $\epsilon > 0$, there exist continuous functions $\phi(x) : \mathcal{X} \to \mathbb{R}^D$ and $\phi(y) : \mathcal{Y} \to \mathbb{R}^D$ with $D \geq 1$ such that $\sup_{x \in \mathcal{X}, y \in \mathcal{Y}} |f(x, y) - \phi(x)^\top \psi(y)| < \varepsilon$.*

*Proof.* We invoke the Stone-Weierstrass theorem (Bass (2013), Theorem 20.44), which implies that the set of functions $\mathcal{T} := \{\sum_{i=1}^{D} \phi_i(x)\psi_i(y) : D \in \mathbb{N}, \forall 1 \leq i \leq D, \phi_i \in \mathcal{C}(\mathcal{X}), \psi_i \in \mathcal{C}(\mathcal{Y})\}$ is dense in $\mathcal{C}(\mathcal{X} \times \mathcal{Y})$ if $\mathcal{T}$ is an algebra that separates points and vanishes at no point. The only non-trivial part is to show that $\mathcal{T}$ is closed under multiplication. Consider $g^{(1)}(x, y) = \sum_{i=1}^{D_1} \phi_i^{(1)}(x)\psi_i^{(1)}(y) \in \mathcal{T}$ and $g^{(2)}(x, y) = \sum_{i=1}^{D_2} \phi_i^{(2)}(x)\psi_i^{(2)}(y) \in \mathcal{T}$. We have $g^{(1)}(x, y)g^{(2)}(x, y) = \sum_{i=1}^{D_1} \sum_{j=1}^{D_2} \phi_i^{(1)}(x)\phi_j^{(2)}(x)\psi_i^{(1)}(y)\psi_j^{(2)}(y)$, where $\phi_i^{(1)}(x)\phi_j^{(2)}(x) \in \mathcal{C}(\mathcal{X})$ and $\psi_i^{(1)}(y)\psi_j^{(2)}(y) \in \mathcal{C}(\mathcal{Y})$ for all $i, j$. Hence, $g^{(1)}(x, y)g^{(2)}(x, y) \in \mathcal{T}$. □

**Theorem C.2** *($\phi(x)^\top \psi(y)$ is a universal approximator of $f(x, y)$).* *Let $\mathcal{X}$ and $\mathcal{Y}$ be compact Hausdorff spaces and $\Phi \subset \mathcal{C}(\mathcal{X})$ and $\Psi \subset \mathcal{C}(\mathcal{Y})$ be dense sets in $\mathcal{C}(\mathcal{X})$ and $\mathcal{C}(\mathcal{Y})$, respectively. Then, $\mathcal{T} := \{\sum_{i=1}^{D} \phi_i(x)\psi_i(y) : D \in \mathbb{N}, \forall 1 \leq i \leq D, \phi_i \in \Phi, \psi_i \in \Psi\}$ is also dense in $\mathcal{C}(\mathcal{X} \times \mathcal{Y})$. In other words, $\phi(x)^\top \psi(y)$ can approximate $f(x, y)$ to arbitrary accuracy if $\phi$ and $\psi$ are modeled with universal approximators (e.g., neural networks) and $D \to \infty$.*

*Proof.* By Lemma C.1, for any $f \in \mathcal{C}(\mathcal{X} \times \mathcal{Y})$ and $\varepsilon > 0$, there exist $D \in \mathbb{N}$, $\phi_i \in \mathcal{C}(\mathcal{X})$, and $\psi_i \in \mathcal{C}(\mathcal{Y})$ for $1 \leq i \leq D$ such that $\sup_{x \in \mathcal{X}, y \in \mathcal{Y}} |f(x, y) - \sum_{i=1}^{D} \phi_i(x)\psi_i(y)| < \varepsilon/3$. Define

$M_y := \sup_{1 \le i \le D, y \in \mathcal{Y}} |\psi_i(y)|$. Since $\Phi$ is dense, for each $1 \le i \le D$, there exists $\tilde{\phi}_i \in \Phi$ such that $\sup_{x \in \mathcal{X}} |\phi_i(x) - \tilde{\phi}_i(x)| < \varepsilon/(3DM_y)$. Define $M_x := \sup_{1 \le i \le D, x \in \mathcal{X}} |\tilde{\phi}_i(x)|$. Similarly, for each $1 \le i \le D$, there exists $\tilde{\psi}_i \in \Psi$ such that $\sup_{y \in \mathcal{Y}} |\psi_i(y) - \tilde{\psi}_i(y)| < \varepsilon/(3DM_x)$. Now, we have

$$\left| f(x,y) - \sum_{i=1}^{D} \tilde{\phi}_i(x)\tilde{\psi}_i(y) \right| \le \left| f(x,y) - \sum_{i=1}^{D} \phi_i(x)\psi_i(y) \right| + \sum_{i=1}^{D} \left| \tilde{\phi}_i(x)\tilde{\psi}_i(y) - \phi_i(x)\psi_i(y) \right| \quad (9)$$

$$< \frac{\varepsilon}{3} + \sum_{i=1}^{D} |\tilde{\phi}_i(x)(\tilde{\psi}_i(y) - \psi_i(y))| + \sum_{i=1}^{D} |(\tilde{\phi}_i(x) - \phi_i(x))\psi_i(y)|$$
$$(10)$$

$$< \frac{\varepsilon}{3} + \frac{\varepsilon}{3} + \frac{\varepsilon}{3} \quad (11)$$

$$= \varepsilon, \quad (12)$$

for any $x \in \mathcal{X}$ and $y \in \mathcal{Y}$. Hence, $\mathcal{T}$ is dense in $\mathcal{C}(\mathcal{X} \times \mathcal{Y})$. □

## C.2  LIPSCHITZ CONSTRAINT UNDER THE TEMPORAL DISTANCE METRIC

**Theorem C.3.** *The following two conditions are equivalent:*

*(a)* $\|\phi(u) - \phi(v)\|_2 \le d_{\text{temp}}(u,v)$ *for all* $u, v \in \mathcal{S}$.

*(b)* $\|\phi(s) - \phi(s')\|_2 \le 1$ *for all* $(s, s') \in \mathcal{S}_{\text{adj}}$.

*Proof.* We first show *(a)* implies *(b)*. Assume *(a)* holds. Consider $(s, s') \in \mathcal{S}_{\text{adj}}$. If $s \ne s'$, by *(a)*, we have $\|\phi(s) - \phi(s')\|_2 \le d_{\text{temp}}(s, s') = 1$. Otherwise, *i.e.*, $s = s'$, $\|\phi(s) - \phi(s')\|_2 = 0 \le 1$. Hence, *(a)* implies *(b)*.

Next, we show *(b)* implies *(a)*. Assume *(b)* holds. Consider $u, v \in \mathcal{S}$. If $d_{\text{temp}}(u,v) = \infty$ (*i.e.*, $v$ is not reachable from $u$), *(a)* holds trivially. Otherwise, let $k$ be $d_{\text{temp}}(u,v)$. By definition, there exists $(s_0 = u, s_1, \ldots, s_{k-1}, s_k = v)$ such that $(s_i, s_{i+1}) \in \mathcal{S}_{\text{adj}}$ for all $0 \le i \le k-1$. Due to the triangle inequality and *(b)*, we have $\|\phi(u) - \phi(v)\|_2 \le \sum_{i=0}^{k-1} \|\phi(s_i) - \phi(s_{i+1})\|_2 \le k = d_{\text{temp}}(u,v)$. Hence, *(b)* implies *(a)*. □

## D  A CONNECTION BETWEEN METRA AND PCA

In this section, we derive a theoretical connection between METRA and principal component analysis (PCA). Recall that the METRA objective can be written as follows:

$$\sup_{\pi, \phi} \mathbb{E}_{p(\tau, z)} \left[ \sum_{t=0}^{T-1} (\phi(s_{t+1}) - \phi(s_t))^\top z \right] = \mathbb{E}_{p(\tau, z)} \left[ \phi(s_T)^\top z \right] \quad (13)$$

$$\text{s.t. } \|\phi(u) - \phi(v)\|_2 \le d_{\text{temp}}(u,v), \quad \forall u, v \in \mathcal{S}, \quad (14)$$

where $d_{\text{temp}}$ denotes the temporal distance between two states. To make a formal connection between METRA and PCA, we consider the following squared variant of the METRA objective in this section.

$$\sup_{\pi, \phi} \mathbb{E}_{p(\tau, z)} \left[ (\phi(s_T)^\top z)^2 \right] \text{ s.t. } \|\phi(u) - \phi(v)\|_2 \le d_{\text{temp}}(u,v), \quad \forall u, v \in \mathcal{S}, \quad (15)$$

which is almost the same as Equation (13) but the objective is now squared. The reason we consider this variant is simply for mathematical convenience.

Next, we introduce the notion of a temporally consistent embedding.

**Definition D.1** (Temporally consistent embedding). We call that an MDP $\mathcal{M}$ admits a temporally consistent embedding if there exists $\psi(s) : \mathcal{S} \to \mathbb{R}^m$ such that

$$d^{\text{temp}}(u,v) = \|\psi(u) - \psi(v)\|_2, \quad \forall u, v \in \mathcal{S}. \quad (16)$$

Intuitively, this states that the temporal distance metric can be embedded into a (potentially very high-dimensional) Euclidean space. We note that $\psi$ is different from $\phi$ in Equation (13), and $\mathbb{R}^m$ can be much higher-dimensional than $\mathcal{Z}$. An example of an MDP that admits a temporally consistent embedding is the PointMass environment: if an agent in $\mathbb{R}^n$ can move in any direction up to a unit speed, $\psi(x) = x$ satisfies $d^{\text{temp}}(u, v) = \|u - v\|_2$ for all $u, v \in \mathbb{R}^n$ (with a slightly generalized notion of temporal distances in continuous time) and thus the MDP admits the temporally consistent embedding of $\psi$. A pixel-based PointMass environment is another example of such an MDP.

Now, we formally derive a connection between squared METRA and PCA. For simplicity, we assume $\mathcal{Z} = \mathbb{R}^d$ and $p(z) = \mathcal{N}(0, \text{I}_d)$, where $\mathcal{N}(0, \text{I}_d)$ denotes the $d$-dimensional isotropic Gaussian distribution. We also assume that $\mathcal{M}$ has a deterministic initial distribution and transition dynamics function, and every state is reachable from the initial state within $T$ steps. We denote the set of $n \times n$ positive definite matrices as $\mathbb{S}^n_{++}$, the operator norm of a matrix $A$ as $\|A\|_{\text{op}}$, and the $m$-dimensional unit $\ell_2$ ball as $\mathbb{B}^m$.

**Theorem D.2** (Linear squared METRA is PCA in the temporal embedding space). *Let $\mathcal{M}$ be an MDP that admits a temporally consistent embedding $\psi : \mathcal{S} \to \mathbb{R}^m$. If $\phi : \mathcal{S} \to \mathcal{Z}$ is a linear mapping from the embedding space, i.e., $\phi(s) = W^\top \psi(s)$ with $W \in \mathbb{R}^{m \times d}$, and the embedding space $\Psi = \{\psi(s) : s \in \mathcal{S}\}$ forms an ellipse, i.e., $\exists A \in \mathbb{S}^m_{++}$ s.t. $\Psi = \{x \in \mathbb{R}^m : x^\top A^{-1} x \leq 1\}$, then $W = [a_1 \ a_2 \ \cdots \ a_d]$ maximizes the squared METRA objective in Equation (15), where $a_1, \ldots, a_d$ are the top-$d$ eigenvectors of $A$.*

*Proof.* Since $\mathcal{M}$ admits a temporally consistent embedding, we have

$$\|\phi(u) - \phi(v)\|_2 \leq d_{\text{temp}}(u, v) \ \forall u, v \in \mathcal{S} \tag{17}$$

$$\Longleftrightarrow \|W^\top(\psi(u) - \psi(v))\|_2 \leq \|\psi(u) - \psi(v)\|_2 \ \forall u, v \in \mathcal{S} \tag{18}$$

$$\Longleftrightarrow \|W^\top(u - v)\|_2 \leq \|u - v\|_2 \ \forall u, v \in \Psi \tag{19}$$

$$\Longleftrightarrow \|W\|_{\text{op}} \leq 1, \tag{20}$$

where we use the fact that $\psi$ is a surjection from $\mathcal{S}$ to $\Psi$ and that $A$ is positive definite. Now, we have

$$= \sup_{\pi, \|W\|_{\text{op}} \leq 1} \mathbb{E}_{p(\tau, z)}[(\phi(s_T)^\top z)^2] \tag{21}$$

$$= \sup_{\pi, \|W\|_{\text{op}} \leq 1} \mathbb{E}_{p(\tau, z)}[(\psi(s_T)^\top W z)^2] \tag{22}$$

$$= \sup_{f : \mathbb{R}^d \to \Psi, \|W\|_{\text{op}} \leq 1} \mathbb{E}_{p(z)}[(f(z)^\top W z)^2] \quad (\because \text{Every state is reachable within } T \text{ steps}) \tag{23}$$

$$= \sup_{g : \mathbb{R}^d \to \mathbb{B}^m, \|W\|_{\text{op}} \leq 1} \mathbb{E}_{p(z)}[(g(z)^\top \sqrt{A} W z)^2] \quad (g(z) = \sqrt{A^{-1}} f(z)) \tag{24}$$

$$= \sup_{\|W\|_{\text{op}} \leq 1} \mathbb{E}_{p(z)}[\sup_{g : \mathbb{R}^d \to \mathbb{B}^m} (g(z)^\top \sqrt{A} W z)^2] \tag{25}$$

$$= \sup_{\|W\|_{\text{op}} \leq 1} \mathbb{E}_{p(z)}[\sup_{\|u\|_2 \leq 1} (u^\top \sqrt{A} W z)^2] \tag{26}$$

$$= \sup_{\|W\|_{\text{op}} \leq 1} \mathbb{E}_{p(z)}[\|\sqrt{A} W z\|_2^2] \quad (\because \text{Dual norm}) \tag{27}$$

$$= \sup_{\|W\|_{\text{op}} \leq 1} \mathbb{E}_{p(z)}[z^\top W^\top A W z] \tag{28}$$

$$= \sup_{\|W\|_{\text{op}} \leq 1} \mathbb{E}_{p(z)}[\text{tr}(zz^\top W^\top A W)] \tag{29}$$

$$= \sup_{\|W\|_{\text{op}} \leq 1} \text{tr}(\mathbb{E}_{p(z)}[zz^\top] W^\top A W) \tag{30}$$

$$= \sup_{\|W\|_{\text{op}} \leq 1} \text{tr}(W W^\top A). \tag{31}$$

Since $W W^\top$ is a positive semidefinite matrix with rank at most $d$ and $\|W\|_{\text{op}} \leq 1$, there exists $d$ eigenvalues $0 \leq \lambda_1, \ldots, \lambda_d \leq 1$ and the corresponding orthonormal eigenvectors $v_1, \ldots, v_d$ such that $W W^\top = \sum_{k=1}^d \lambda_k v_k v_k^\top$. Hence, $\text{tr}(W W^\top A) = \sum_{k=1}^d \lambda_k v_k^\top A v_k$, and to maximize this, we

must set $\lambda_1 = \cdots = \lambda_d = 1$ as $A$ is positive definite. The remaining problem is to find $d$ orthonormal vectors $v_1, \ldots, v_d$ that maximize $\sum_{k=1}^{d} v_k^\top A v_k$. By the Ky Fan's maximum principle (Bhatia, 2013), its solution is given as the $d$ eigenvectors corresponding to the $d$ largest eigenvalues of $A$. Therefore, $W = [a_1 \ a_2 \ \cdots \ a_d]$, where $a_1, \ldots, a_d$ are the top-$d$ principal components of $A$, maximizes the squared METRA objective in Equation (15). □

Theorem D.2 states that linear squared METRA is equivalent to PCA in the temporal embedding space. In practice, however, $\phi$ can be nonlinear, the shape of $\Psi$ can be arbitrary, and the MDP may not admit any temporally consistent embeddings. Nonetheless, this theoretical connection hints at the intuition that the METRA objective encourages the agent to span the largest "temporal" manifolds in the state space, given the limited capacity of $\mathcal{Z}$.

## E    CONNECTIONS BETWEEN WDM AND DIAYN, DADS, AND CIC

In this section, we describe connections between our WDM objectives (either $I_\mathcal{W}(S; Z)$ or $I_\mathcal{W}(S_T; Z)$) and previous mutual information skill learning methods, DIAYN (Eysenbach et al., 2019a), DADS (Sharma et al., 2020), and CIC (Laskin et al., 2022). Recall that the $I_\mathcal{W}(S; Z)$ objective (Equation (4)) maximizes

$$\sum_{t=0}^{T-1} \left( \mathbb{E}_{p(\tau,z)}[\phi_L(s_t)^\top \psi_L(z)] - \mathbb{E}_{p(\tau)}[\phi_L(s_t)]^\top \mathbb{E}_{p(z)}[\psi_L(z)] \right), \tag{32}$$

and the $I_\mathcal{W}(S_T; Z)$ objective (Equation (6)) maximizes

$$\sum_{t=0}^{T-1} \left( \mathbb{E}_{p(\tau,z)}[(\phi_L(s_{t+1}) - \phi_L(s_t))^\top \psi_L(z)] - \mathbb{E}_{p(\tau)}[\phi_L(s_{t+1}) - \phi_L(s_t)]^\top \mathbb{E}_{p(z)}[\psi_L(z)] \right), \tag{33}$$

where we use the notations $\phi_L$ and $\psi_L$ to denote that they are Lipschitz constrained. By simplifying Equation (32) or Equation (33) in three different ways, we will show that we can obtain "Wasserstein counterparts" of DIAYN, DADS, and CIC. For simplicity, we assume $p(z) = \mathcal{N}(0, \mathrm{I})$, where $\mathcal{N}(0, \mathrm{I})$ denotes the standard Gaussian distribution.

### E.1    DIAYN

If we set $\psi_L(z) = z$ in Equation (32), we get

$$r_t = \phi_L(s_t)^\top z. \tag{34}$$

This is analogous to DIAYN (Eysenbach et al., 2019a), which maximizes

$$I(S; Z) = -H(Z|S) + H(Z) \tag{35}$$

$$\gtrsim \mathbb{E}_{p(\tau,z)} \left[ \sum_{t=0}^{T-1} \log q(z|s_t) \right] \tag{36}$$

$$\simeq \mathbb{E}_{p(\tau,z)} \left[ \sum_{t=0}^{T-1} -\|z - \phi(s_t)\|_2^2 \right], \tag{37}$$

$$r_t^{\text{DIAYN}} = -\|\phi(s_t) - z\|_2^2, \tag{38}$$

where '$\gtrsim$' and '$\simeq$' respectively denote '$>$' and '$=$' up to constant scaling or shifting, and we assume that the variational distribution $q(z|s)$ is modeled as $\mathcal{N}(\phi(s), \mathrm{I})$. By comparing Equation (34) and Equation (38), we can see that Equation (34) can be viewed as a Lipschitz, inner-product variant of DIAYN. This analogy will become clearer later.

### E.2    DADS

If we set $\phi_L(s) = s$ in Equation (33), we get

$$r_t = (s_{t+1} - s_t)^\top \psi_L(z) - (s_{t+1} - s_t)^\top \mathbb{E}_{p(z)}[\psi_L(z)] \tag{39}$$

$$\approx (s_{t+1} - s_t)^\top \psi_L(z) - \frac{1}{L} \sum_{i=1}^{L} (s_{t+1} - s_t)^\top \psi_L(z_i), \tag{40}$$

where we use $L$ independent samples from $\mathcal{N}(0, I)$, $z_1, z_2, \ldots, z_L$, to approximate the expectation. This is analogous to DADS (Sharma et al., 2020), which maximizes:

$$I(S'; Z|S) = -H(S'|S, Z) + H(S'|S) \tag{41}$$

$$\gtrsim \mathbb{E}_{p(\tau,z)}\left[\sum_{t=0}^{T-1} \log q(s_{t+1}|s_t, z) - \log p(s_{t+1}|s_t)\right] \tag{42}$$

$$\approx \mathbb{E}_{p(\tau,z)}\left[\sum_{t=0}^{T-1}\left(\log q(s_{t+1}|s_t, z) - \frac{1}{L}\sum_{i=1}^{L} \log q(s_{t+1}|s_t, z_i)\right)\right], \tag{43}$$

$$r_t^{\text{DADS}} = -\|(s_{t+1} - s_t) - \psi(s_t, z)\|_2^2 + \frac{1}{L}\sum_{i=1}^{L}\|(s_{t+1} - s_t) - \psi(s_t, z)\|_2^2, \tag{44}$$

where we assume that the variational distribution $q(s'|s, z)$ is modeled as $q(s' - s|s, z) = \mathcal{N}(\psi(s, z), I)$, as in the original implementation (Sharma et al., 2020). We also use the same sample-based approximation as Equation (40). Note that the same analogy also holds between Equation (40) and Equation (44) (*i.e.*, Equation (40) is a Lipschitz, inner-product variant of DADS).

### E.3 CIC

If we do not simplify $\phi_L$ or $\psi_L$ in Equation (32), we get

$$r_t = \phi_L(s_t)^\top \psi_L(z) - \phi_L(s_t)^\top \mathbb{E}_{p(z)}[\psi_L(z)] \tag{45}$$

$$\approx \phi_L(s_t)^\top \psi_L(z) - \frac{1}{L}\sum_{i=1}^{L} \phi_L(s_t)^\top \psi_L(z_i), \tag{46}$$

where we use the same sample-based approximation as Equation (40). By Jensen's inequality, Equation (46) can be lower-bounded by

$$\phi_L(s_t)^\top \psi_L(z) - \log \frac{1}{L}\sum_{i=1}^{L} \exp\left(\phi_L(s_t)^\top \psi_L(z_i)\right), \tag{47}$$

as in WPC (Ozair et al., 2019). This is analogous to CIC (Laskin et al., 2022), which estimates the MI via noise contrastive estimation (Gutmann & Hyvärinen, 2010; van den Oord et al., 2018; Poole et al., 2019):

$$I(S; Z) \gtrsim \mathbb{E}_{p(\tau,z)}\left[\sum_{t=0}^{T-1}\left(\phi(s_t)^\top \psi(z) - \log \frac{1}{L}\sum_{i=1}^{L} \exp\left(\phi(s_t)^\top \psi(z_i)\right)\right)\right], \tag{48}$$

$$r_t^{\text{CIC}} = \phi(s_t)^\top \psi(z) - \log \frac{1}{L}\sum_{i=1}^{L} \exp\left(\phi(s_t)^\top \psi(z_i)\right). \tag{49}$$

Note that Equation (47) can be viewed as a Lipschitz variant of CIC (Equation (49)).

In this work, we use the $\psi_L(z) = z$ simplification with Equation (33) (*i.e.*, Equation (7)), as we found this variant to work well while being simple, but we believe exploring these other variants is an interesting future research direction. In particular, given that Equation (47) resembles the standard contrastive learning formulation, combining this (more general) objective with existing contrastive learning techniques may lead to another highly scalable unsupervised RL method, which we leave for future work.

## F ADDITIONAL RESULTS

### F.1 FULL QUALITATIVE RESULTS

Figure 11 shows the complete qualitative results of behaviors discovered by METRA on state-based Ant and HalfCheetah, and pixel-based Quadruped and Humanoid (8 seeds for each environment). We use 2-D skills for Ant and Humanoid, 4-D skills for Quadruped, and 16 discrete skills for HalfCheetah. The full qualitative results suggest that METRA discovers diverse locomotion behaviors regardless of the random seed.

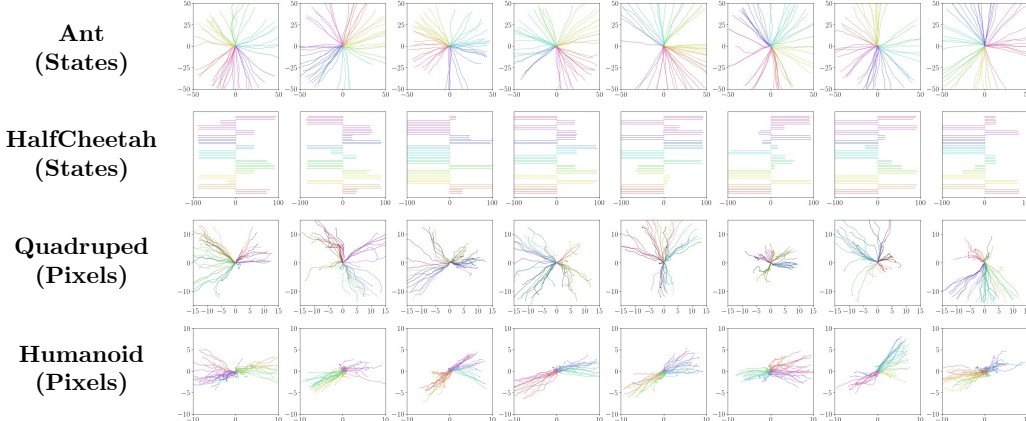

Figure 11: **Full qualitative results of METRA (8 seeds).** METRA learns diverse locomotion behaviors regardless of the random seed.

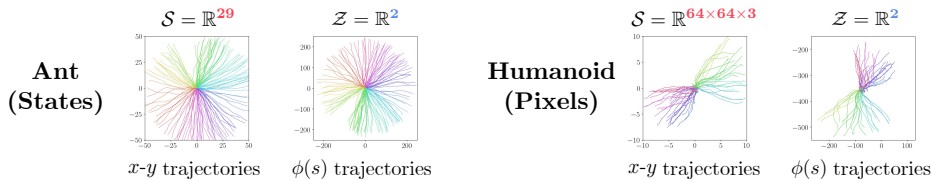

Figure 12: **Latent space visualization.** METRA learns to capture $x$-$y$ coordinates in two-dimensional latent spaces in both state-based Ant and pixel-based Humanoid, as they are the most temporally spread-out dimensions in the state space. We note that, with a higher-dimensional latent space (especially when $\mathcal{Z}$ is discrete), METRA not only learns locomotion skills but also captures more diverse behaviors, as shown in the Cheetah and Kitchen videos on our project page.

## F.2  LATENT SPACE VISUALIZATION

METRA simultaneously learns both the skill policy $\pi(a|s,z)$ and the representation function $\phi(s)$, to find the most "temporally spread-out" manifold in the state space. We train METRA on state-based Ant and pixel-based Humanoid with 2-D continuous latent spaces $\mathcal{Z}$, and visualize the learned latent space by plotting $\phi(s)$ trajectories in Figure 12. Since the $x$-$y$ plane corresponds to the most temporally "important" manifold in both environments, METRA learns to capture the $x$-$y$ coordinates in two-dimensional $\phi$, regardless of the input representations (note that Humanoid is pixel-based). We also note that, with a higher-dimensional latent space (especially when $\mathcal{Z}$ is discrete), METRA not only learns locomotion skills but also captures more diverse, non-linear behaviors, as shown in the Cheetah and Kitchen videos on our project page.

## F.3  ABLATION STUDY OF LATENT SPACE SIZES

To demonstrate how the size of the latent space $\mathcal{Z}$ affects skill learning, we train METRA with 1-D, 2-D, and 4-D continuous skills and 2, 4, 8, 16, and 24 discrete skills on Ant and HalfCheetah. Figure 13 compares skills learned with different latent space sizes, which suggests that the diversity of skill generally increases as the capacity of $\mathcal{Z}$ grows.

## F.4  ADDITIONAL BASELINES

In the main paper, we compare METRA with 11 previous unsupervised exploration and unsupervised skill discovery methods. In this section, we additionally compare METRA with DGPO (Chen et al., 2024), a method that aims to find diverse behaviors that maximize a task reward (Kumar et al., 2020; Zhou et al., 2022; Zahavy et al., 2023a;b; Chen et al., 2024). Since we consider a controlled Markov process without external rewards, we use only the intrinsic reward part of DGPO for

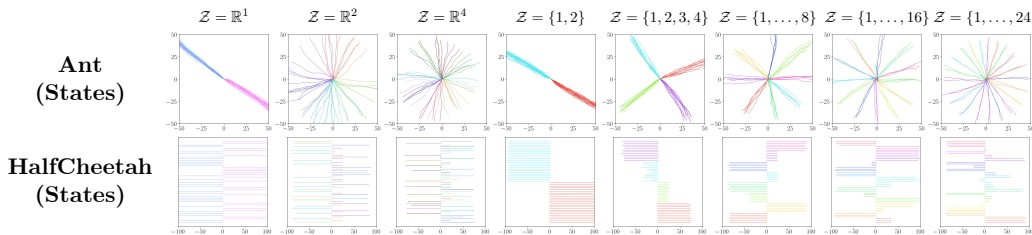

Figure 13: **Skills learned with different latent space sizes.** Since METRA maximizes state coverage under the capacity of the latent space $\mathcal{Z}$, skills become more diverse as the capacity of $\mathcal{Z}$ grows.

Table 1: **Comparison with DGPO.** We compare METRA with an additional baseline for discrete skill learning, DGPO (Chen et al., 2024). METRA exhibits the best state coverage in both Ant and HalfCheetah. We use 4 random seeds and '$\pm$' denotes standard deviations.

| Environment (Metric) | DIAYN | DGPO | METRA (ours) |
|---|---|---|---|
| HalfCheetah (policy state coverage) | $6.75 _{\pm 2.22}$ | $6.75 _{\pm 2.06}$ | $\mathbf{186.75} _{\pm 16.21}$ |
| HalfCheetah (total state coverage) | $19.50 _{\pm 3.87}$ | $22.25 _{\pm 5.85}$ | $\mathbf{177.75} _{\pm 17.10}$ |
| Ant (policy state coverage) | $11.25 _{\pm 5.44}$ | $7.00 _{\pm 3.83}$ | $\mathbf{1387.75} _{\pm 77.38}$ |
| Ant (total state coverage) | $107.75 _{\pm 17.00}$ | $121.50 _{\pm 4.36}$ | $\mathbf{6313.25} _{\pm 747.92}$ |

comparison:

$$r_t^{\text{DGPO}} = \min_{z' \in \mathcal{Z}, z' \neq z} \log \frac{q(z|s_{t+1})}{q(z|s_{t+1}) + q(z'|s_{t+1})}, \tag{50}$$

where $q$ is a skill discriminator (Eysenbach et al., 2019a) and DGPO assumes that $\mathcal{Z}$ is a discrete space. Intuitively, this objective encourages each behavior to be maximally different from the most similar other behavior.

Table 1 presents the comparison results on HalfCheetah and Ant, where we train DIAYN, DGPO, and METRA with 16 discrete skills for 10000 epochs (16M steps). Even though DGPO maximizes "worst-case" diversity (Equation (50)), it still maximizes a *metric-agnostic* KL divergence between different skills (Chen et al., 2024), which leads to limited state coverage, as in DIAYN. In contrast, METRA maximizes a *metric-aware* Wasserstein distance and thus shows significantly better state coverage.

# G EXPERIMENTAL DETAILS

We implement METRA on top of the publicly available LSD codebase (Park et al., 2022). Our implementation is available at https://github.com/seohongpark/METRA. For unsupervised skill discovery methods, we implement LSD (Park et al., 2022), CIC (Laskin et al., 2022), DIAYN (Eysenbach et al., 2019a), and DADS (Sharma et al., 2020) on the same codebase as METRA. For six exploration methods, ICM (Pathak et al., 2017), LBS (Mazzaglia et al., 2022), RND (Burda et al., 2019), APT (Liu & Abbeel, 2021b), APS (Liu & Abbeel, 2021a), and Plan2Explore (Sekar et al., 2020) (or Disagreemeent (Pathak et al., 2019)), we use the original implementations by Laskin et al. (2021) for state-based environments and the Dreamer (Hafner et al., 2020) variants by Rajeswar et al. (2023) for pixel-based environments. For LEXA (Mendonca et al., 2021) in Section 5.3, we use the original implementation by Mendonca et al. (2021). We run our experiments on an internal cluster consisting of A5000 GPUs. Each run in Section 5.3 takes no more than 24 hours.

## G.1 ENVIRONMENTS

**Benchmark environments.** For state-based environments, we use the same MuJoCo HalfCheetah and Ant environments (Todorov et al., 2012; Brockman et al., 2016) as previous work (Sharma et al., 2020; Park et al., 2022; 2023b). HalfCheetah has an 18-dimensional state space and Ant has a 29-dimensional state space. For pixel-based environments, we use pixel-based Quadruped and Humanoid from the DeepMind Control Suite (Tassa et al., 2018) and a pixel-based version of

Kitchen by Gupta et al. (2019); Mendonca et al. (2021). In DMC locomotion environments, we use gradient-colored floors to allow the agent to infer its location from pixels, similarly to Hafner et al. (2022); Park et al. (2023a). In Kitchen, we use the same camera setting as LEXA (Mendonca et al., 2021). Pixel-based environments have an observation space of $64 \times 64 \times 3$, and we do not use any proprioceptive state information. The episode length is 200 for Ant and HalfCheetah, 400 for Quadruped and Humanoid, and 50 for Kitchen. We use an action repeat of 2 for pixel-based Quadruped and Humanoid, following Mendonca et al. (2021). In our experiments, we do not use any prior knowledge or supervision, such as the $x$-$y$ prior (Eysenbach et al., 2019a; Sharma et al., 2020), or early termination (Park et al., 2022).

**Metrics.** For the state coverage metric in locomotion environments, we count the number of $1 \times 1$-sized $x$-$y$ bins (Ant, Quadruped, and Humanoid) or 1-sized $x$ bins (HalfCheetah) that are occupied by any of the target trajectories. In Kitchen, we count the number of pre-defined tasks achieved by any of the target trajectories, where we use the same six pre-defined tasks as Mendonca et al. (2021): Kettle (K), Microwave (M), Light Switch (LS), Hinge Cabinet (HC), Slide Cabinet (SC), and Bottom Burner (BB). Each of the three types of coverage metrics, policy state coverage (Figures 5 and 7), queue state coverage (Figure 7), and total state coverage (Figure 7), uses different target trajectories. Policy state coverage, which is mainly for skill discovery methods, is computed by 48 deterministic trajectories with 48 randomly sample skills at the current epoch. Queue state coverage is computed by the most recent 100000 training trajectories up to the current epoch. Total state coverage is computed by the entire training trajectories up to the current epoch.

**Downstream tasks.** For quantitative comparison of skill discovery methods (Figure 6), we use five downstream tasks, AntMultiGoals, HalfCheetahGoal, HalfCheetahHurdle, QuadrupedGoal, and HumanoidGoal, mostly following the prior work (Park et al., 2022). In HalfCheetahGoal, QuadrupedGoal, and HumanoidGoal, the task is to reach a target goal (within a radius of 3) randomly sampled from $[-100, 100]$, $[-7.5, 7.5]^2$, and $[-5, 5]^2$, respectively. The agent receives a reward of 10 when it reaches the goal. In AntMultiGoals, the task is to reach four target goals (within a radius of 3), where each goal is randomly sampled from $[s_x - 7.5, s_x + 7.5] \times [s_y - 7.5, s_y + 7.5]$, where $(s_x, s_y)$ is the agent's current $x$-$y$ position. The agent receives a reward of 2.5 whenever it reaches the goal. A new goal is sampled when the agent either reaches the previous goal or fails to reach it within 50 steps. In HalfCheetahHurdle (Qureshi et al., 2020), the task is to jump over multiple hurdles. The agent receives a reward of 1 whenever it jumps over a hurdle. The episode length is 200 for state-based environments and 400 for pixel-based environments.

For quantitative comparison with LEXA (Figure 8), we use five goal-conditioned tasks. In locomotion environments, goals are randomly sampled from $[-100, 100]$ (HalfCheetah), $[-50, 50]^2$ (Ant), $[-15, 15]^2$ (Quadruped), or $[-10, 10]^2$ (Humanoid). We provide the full state as a goal $g$, whose dimensionality is 18 for HalfCheetah, 29 for Ant, and $64 \times 64 \times 3$ for pixel-based Quadruped and Humanoid. In Kitchen, we use the same six (single-task) goal images and tasks as Mendonca et al. (2021). We measure the distance between the goal and the final state in locomotion environments and the number of successful tasks in Kitchen.

### G.2 IMPLEMENTATION DETAILS

**Unsupervised skill discovery methods.** For skill discovery methods, we use 2-D continuous skills for Ant and Humanoid, 4-D continuous skills for Quadruped, 16 discrete skills for HalfCheetah, and 24 discrete skills for Kitchen, where continuous skills are sampled from the standard Gaussian distribution, and discrete skills are uniformly sampled from the set of zero-centered one-hot vectors (Park et al., 2022). METRA and LSD use normalized vectors (*i.e.*, $z/\|z\|_2$) for continuous skills, as their objectives are invariant to the magnitude of $z$. For CIC, we use 64-D continuous skills for all environments, following the original suggestion (Laskin et al., 2022), and we found that using 64-D skills for CIC leads to better state coverage than using 2-D or 4-D skills. We present the full list of hyperparameters used for skill discovery methods in Table 2.

**Unsupervised exploration methods.** For unsupervised exploration methods and LEXA, we use the original implementations and hyperparameters (Laskin et al., 2021; Mendonca et al., 2021; Rajeswar et al., 2023). For LEXA's goal-conditioned policy (achiever), we test both the temporal distance and cosine distance variants and use the former as it leads to better performance.

Table 2: Hyperparameters for unsupervised skill discovery methods.

| Hyperparameter | Value |
|---|---|
| Learning rate | 0.0001 |
| Optimizer | Adam (Kingma & Ba, 2015) |
| # episodes per epoch | 8 |
| # gradient steps per epoch | 200 (Quadruped, Humanoid), 100 (Kitchen), 50 (Ant, HalfCheetah) |
| Minibatch size | 256 |
| Discount factor $\gamma$ | 0.99 |
| Replay buffer size | $10^6$ (Ant, HalfCheetah), $10^5$ (Kitchen), $3 \times 10^5$ (Quadruped, Humanoid) |
| Encoder | CNN (LeCun et al., 1989) |
| # hidden layers | 2 |
| # hidden units per layer | 1024 |
| Target network smoothing coefficient | 0.995 |
| Entropy coefficient | 0.01 (Kitchen), auto-adjust (Haarnoja et al., 2018b) (others) |
| METRA $\varepsilon$ | $10^{-3}$ |
| METRA initial $\lambda$ | 30 |

Table 3: Hyperparameters for PPO high-level controllers.

| Hyperparameter | Value |
|---|---|
| # episodes per epoch | 64 |
| # gradient steps per episode | 10 |
| Minibatch size | 256 |
| Entropy coefficient | 0.01 |
| GAE $\lambda$ (Schulman et al., 2016) | 0.95 |
| PPO clip threshold $\epsilon$ | 0.2 |

**High-level controllers for downstream tasks.** In Figure 6, we evaluate learned skills on downstream tasks by training a high-level controller $\pi^h(z|s, s^{\text{task}})$ that selects a skill every $K = 25$ (Ant and HalfCheetah) or $K = 50$ (Quadruped and Humanoid) environment steps, where $s^{\text{task}}$ denotes the task-specific information: the goal position ('-Goal' or '-MultiGoals' tasks) or the next hurdle position and distance (HalfCheetahHurdle). At every $K$ steps, the high-level policy selects a skill $z$, and then the pre-trained low-level skill policy $\pi(a|s, z)$ executes the same $z$ for $K$ steps. We train high-level controllers with PPO (Schulman et al., 2017) for discrete skills and SAC (Haarnoja et al., 2018a) for continuous skills. For SAC, we use the same hyperparameters as unsupervised skill discovery methods (Table 2), and we present the full list of PPO-specific hyperparameters in Table 3.

**Zero-shot goal-conditioned RL.** In Figure 8, we evaluate the zero-shot performances of METRA, LSD, DIAYN, and LEXA on goal-conditioned downstream tasks. METRA and LSD use the procedure described in Section 4.2 to select skills. We re-compute $z$ every step for locomotion environments, but in Kitchen, we use the same $z$ selected at the first step throughout the episode, as we find that this leads to better performance. DIAYN chooses $z$ based on the output of the skill discriminator at the goal state (*i.e.*, $q(z|g)$, where $q$ denotes the skill discriminator of DIAYN). LEXA uses the goal-conditioned policy (achiever), $\pi(a|s, g)$.

