# OpenReview forum: "METRA: Scalable Unsupervised RL with Metric-Aware Abstraction"
_ICLR.cc/2024/Conference — ICLR 2024 oral_

### Official Review · Reviewer_YkQo · 2023-10-17

**Soundness:** 3 good
**Presentation:** 3 good
**Contribution:** 2 fair
**Rating:** 8
**Confidence:** 3

**Summary:**

This work introduces an innovative algorithm designed to discover approximate state-covering behaviors in a task-agnostic manner for reinforcement learning agents. Building upon existing skill-learning techniques that use the mutual information between skills and states to learn distinguishable and state-covering skills, this work introduces a modification to the objective. The modification involves constraining the latent space of the skills to prioritize temporally compact representations. This novel approach ensures that the skills are estimated within a manifold preserving the temporal properties of the state space, allowing for maximization of spread-out trajectories over time. The incorporation of a metric, specifically temporal distance, grounds the latent space to be compact, distinguishing it from previous metric-agnostic alternatives relying solely on mutual information. The study employs a "Wasserstein variant" of the mutual information objective, adeptly modified to facilitate tractable optimization and online learning. One of the key advantages of pre-training RL agents using this objective lies in its scalability, enabling the learning of a compact skill space even from high-dimensional observations, such as images, where prior methods faced challenges in scalability. Moreover, when training a hierarchical controller to employ these learned skills, METRA demonstrates superior downstream task performance in previously unsolved benchmarks. In essence, the paper makes a significant contribution by introducing an effective new algorithm and strengthens its credibility by providing essential theoretical underpinnings for its design.

**Strengths:**

This work builds upon an extensive body of research focused on skill discovery and learning, leveraging variants of mutual information to guide the pre-training process. This work tackles a limitation inherent in the existing MI objective, which is metric-agnostic, and hence does not directly incentivize skills to maximize an explicit metric for state coverage. The proposed modification facilitates the operation of skills within a temporally compact space, wherein the maximization of diversity aligns with the adoption of state-covering behaviors. The incorporation of a well-defined theoretical formulation enhances the paper's credibility, providing a strong foundation for the proposed objective.

The key contribution of the proposed algorithm is its scalability, particularly when dealing with high-dimensional observations - an ongoing challenge in the broader field of reinforcement learning. By addressing this limitation, the paper contributes significantly to the unsupervised RL domain. This work presents the development of an algorithm that stands as a substantial and impactful contribution to the field of unsupervised RL, and that is clearly theoretically justified.

**Weaknesses:**

One weakness of the paper is the absence of a comparative analysis of the proposed algorithm's performance in alternative benchmarks, especially those where purely exploratory algorithms like RND have demonstrated exceptional results (e.g., in Atari). Specifically, METRA's performance remains unclear in settings such as discrete control or, more significantly, in stochastic environments. An important aspect that is missing is a demonstration of how the temporally compact space learned by METRA enables reward-free pre-training in these challenging scenarios. Addressing these points would significantly enhance the paper's impact, making it a more substantial and comprehensive contribution to the field.

The paper iteratively simplifies the proposed objective to enable tractable optimization. However, it remains unclear if these modifications impact the performance of the algorithm. An ablation study of these could also provide insightful details of the proposed objective. (e.g. if training with the formulation that requires N rollouts for each latent, would the obtained skills be more diverse?)

**Questions:**

The temporal distance is a very natural choice for a distance metric for reinforcement learning. However, are there any other metrics that are promising for their applications to the reinforcement learning setting in the METRA framework? Were these considered for this work?

METRA forces the latent variables to maintain linear relationships in the latent space. Although this allows for non-linear policies in the state space, can the latter limit the diversity of the learned skills in other challenging environments? (e.g. stochastic, multi-agent). Could METRA be modified to model non-linear skills even in the latent space?

---

> ### Author Response · Authors · 2023-11-14
>
> We thank the reviewer for the thorough review and constructive feedback about this work. Below, we present new ablation study results and provide answers to the questions. We believe that these changes strengthen the paper, and welcome additional suggestions for further improving the work.
>
> * **“Could METRA be modified to model non-linear skills even in the latent space?”**, **Ablation study on approximated objectives**
>
> Yes, it is possible to modify METRA to model non-linear skills in the latent space by using the *full* $\psi(z)$ form (instead of simplifying it with $\psi(z) = z$ in Eq. 7), and we discuss this full version of METRA in detail in Appendix E.3. Below, we empirically compare METRA with the original version before the $\psi(z)=z$ simplification (Eq. 6). We test two different dimensionalities for $\psi$ ($D=2$ and $D=16$).
>
> | Environment | DIAYN | DADS | METRA (full, $D=2$) | METRA (full, $D=16$) | METRA (simplified) |
> |---|---|---|---|---|---|
> | HalfCheetah | $2.00 \pm 0.00$ | $2.75 \pm 0.96$ | $\mathbf{183.50} \pm 10.85$ | $155.75 \pm 4.35$ | $171.75 \pm 1.71$ |
> | Ant | $4.75 \pm 0.96$ | $7.25 \pm 1.71$ | $\mathbf{2398.50} \pm 51.00$ | $2209.00 \pm 37.21$ | $2307.25 \pm 150.14$ |
>
> The table above shows the results (policy state coverage) on Ant and HalfCheetah (at epoch 10000, 4 seeds each). In these environments, while both the simplified version (with $\psi(z) = z$) and the full version (without setting $\psi(z) = z$) perform similarly, the setting with $D=16$ exhibits slightly slower learning due to the added complexity.
>
> In general, we believe using the full version of METRA could lead to more diverse skills in highly complex environments, as there is less restriction on latent behaviors (though it could be more computationally expensive or relatively harder to optimize). Another interesting aspect of the full METRA objective is that it resembles contrastive learning (Appendix E.3). As such, combining the full version of METRA with contrastive learning techniques may further help scale up the idea of METRA to more complex or even real-world environments, which we leave for future work.
>
> * **“Are there any other metrics that are promising for their applications to the reinforcement learning setting in the METRA framework?”**
>
> As the reviewer pointed out, it is possible to combine our WDM objective (Eq. 3) with distance metrics other than temporal distances. For example, as mentioned in Sec 4.2, LSD (a previous unsupervised skill discovery method) can be viewed as a special case of WDM with the *Euclidean* distance metric (though this is not as universally applicable as the temporal distance metric). Another potentially interesting metric is the bisimulation metric [1, 2], which measures the similarity between states based on both environment dynamics *and* a reward function. Although this metric requires a pre-defined reward function(s), it may serve as a way to inject human priors about tasks into unsupervised learning of skills. We believe considering and comparing various distance metrics (or quasimetrics) is an interesting future research direction.

---

> > ### Author Response · Authors · 2023-11-14
> >
> > * **METRA in alternative benchmarks (e.g., Atari games)**
> >
> > Thank you for the suggestion. While we agree that evaluating METRA on more diverse benchmarks like Atari games would further strengthen the paper, we would like to note that we have included a large fraction of the most widely used benchmarks for unsupervised RL and unsupervised skill discovery [3, 4, 5, 6, 7, 8, 9, 10, 11, 12, 13], including both locomotion and manipulation environments. Also, please understand that adapting the current METRA implementation to Atari games requires a considerable amount of time and work as they require a very different RL backbone (other than SAC) due to the difference in action spaces. (We also tried to find an existing unsupervised skill discovery implementation for Atari, but most prior methods do not evaluate on Atari games, and the only available [repository](https://github.com/mklissa/dceo) we found seems to require 5-7 days of training.) That being said, we believe further extending the idea behind METRA to other types of environments is an exciting future research direction.
> >
> > As the reviewer pointed out, pure exploration approaches (e.g., RND) achieve decent performance in Atari games (sometimes in combination with extrinsic rewards), as games are often *intentionally* designed for the player to seek novel states or scenes. At the same time, however, pure exploration methods often fail in complex robotics environments (Fig. 3), where it is infeasible to visit every possible state. Having not evaluated METRA on Atari games, we cannot claim that METRA is better than pure exploration approaches on the Atari benchmark. However, we do believe that the idea behind METRA has better potential scalability in principle, as it does not attempt to cover every possible state (and as empirically shown in our experiments), unlike pure exploration methods. We hope that METRA provides a complementary aspect to existing unsupervised RL approaches, toward truly scalable unsupervised RL. We have revised the limitation section regarding this point in the updated manuscript (Appendix A).
> >
> >
> > We thank the reviewer again for the helpful feedback and please let us know if there are any additional concerns or questions.
> >
> > [1] Ferns et al., Metrics for finite Markov decision processes (2004).
> >
> > [2] Castro et al., Scalable methods for computing state similarity in deterministic Markov decision processes (2020).
> >
> > [3] Eysenbach et al., Diversity is all you need: Learning skills without a reward function (2019).
> >
> > [4] Sharma et al., Dynamics-aware unsupervised discovery of skills (2020).
> >
> > [5] Mendonca et al., Discovering and achieving goals via world models (2021).
> >
> > [6] Laskin et al., Urlb: Unsupervised reinforcement learning benchmark (2021).
> >
> > [7] He et al., Wasserstein unsupervised reinforcement learning (2022).
> >
> > [8] Park et al., Lipschitz-constrained unsupervised skill discovery (2022).
> >
> > [9] Zhao et al., A mixture of surprises for unsupervised reinforcement learning (2022).
> >
> > [10] Shafiullah and Pinto, One after another: Learning incremental skills for a changing world (2022).
> >
> > [11] Laskin et al., Unsupervised reinforcement learning with contrastive intrinsic control (2022).
> >
> > [12] Park et al., Controllability-aware unsupervised skill discovery (2023).
> >
> > [13] Yang et al., Behavior contrastive learning for unsupervised skill discovery (2023).

---

> > > ### Comment · Reviewer_YkQo · 2023-11-14
> > >
> > > Thank you for addressing my points. I understand the technical difficulties of implementing works like METRA across different RL algorithms and environments with different specifications (discrete/continuous action spaces). While it is true that the environments used in this paper follow the typical literature in unsupervised RL and skill discovery, it seems that scaling these methods to work from pixel-based observations was one of the last missing components for Mujoco-like tasks. My point remains the same, as I believe that this type of work should aim to also learn diverse skills in more complicated settings (e.g. stochasticity, long-horizon, multi-agent). These results would further develop unsupervised RL towards a generally useful setting.
> > >
> > > Overall, I maintain my evaluation and suggest accepting this paper, but encourage the authors to further push the boundaries of what unsupervised RL is currently capable of in order to broaden the contribution of works like METRA. Specifically, because I believe that you could obtain great results in more interesting environments (e.g. Crafter, Montezuma, Minecraft...)

---

### Official Review · Reviewer_N85D · 2023-10-28

**Soundness:** 3 good
**Presentation:** 3 good
**Contribution:** 3 good
**Rating:** 6
**Confidence:** 3

**Summary:**

This work proposes a novel objective for learning diverse skills in unsupervised skill discovery. In particular, this objective can enforce good policy coverage and is scalable to high-dimensional environments. The authors theoretically analyze the learning process of the Wasserstein dependency measure, and the analysis is accompanied by convincing experiments. The experiments and theoretical analysis show the effectiveness of the proposed approach.

**Strengths:**

1. This paper is well-structured. The authors first analyze the common limitations of existing unsupervised RL approaches and then provide solid theoretical and empirical evidence to show why and how the proposed method works, making this paper understandable.

2. Experiments are well-described and highly reproducible. Experiments have good coverage. The selection of baselines and environments is reasonable and convincing.

**Weaknesses:**

There are no significant weaknesses in this paper. The theoretical explanations of why choosing WDM as the objective might be a little complicated for readers lacking corresponding background. Some explicit examples or pictures may help.

**Questions:**

1. Does METRA have the potential to be applied to environments with temporal dependencies (NetHack, MineDojo, ...)?

2. If we apply METRA to a traditional RL setting with external rewards, how would the discovered skills help in finding task-specific policies?

---

> ### Author Response · Authors · 2023-11-14
>
> We thank the reviewer for the thorough review and constructive feedback about this work. Please find our answers to the questions below. We welcome additional suggestions for further improving the work.
>
> * **“If we apply METRA to a traditional RL setting with external rewards, how would the discovered skills help in finding task-specific policies?”**
>
> In Section 5, we have (already) shown two ways to utilize skills learned by METRA for downstream tasks. First, we train hierarchical high-level policies $\pi^h(z|s)$ that selects skills as temporally extended actions. In Fig. 6, we show that the high-level policies (trained on fixed METRA skills) effectively solve challenging downstream tasks defined by external rewards (e.g., jumping over multiple hurdles, navigating through multiple goals, etc.). Second, specifically for *goal-reaching* tasks, METRA provides a very simple way to directly select a skill in a *zero-shot* manner (Section 4.2). We empirically demonstrate this capability of METRA in Fig. 8, which shows that METRA achieves the best zero-shot goal-reaching performance.
>
> Moreover, while we did not mention in the paper, we can also combine METRA with *successor features* by noting that the METRA reward $(\phi(s’) - \phi(s))^\top z$ can be interpreted as the inner product between the feature vector $\tilde \phi(s, a, s’) = \phi(s’) - \phi(s)$ and the task vector $z$. Based on this interpretation, we can compute via linear regression the most suitable skill $z$ for *any arbitrary reward function* in a zero-shot manner, similarly to successor feature methods. We are currently working on this as a follow-up to METRA.
>
> * **“Does METRA have the potential to be applied to environments with temporal dependencies (NetHack, MineDojo, ...)?”**
>
> METRA can be applied to any fully observable MDP in principle. However, METRA in its current form does not particularly deal with partially observable environments (e.g., Minecraft), like many prior works in unsupervised skill discovery [1, 2, 3, 4]. We believe this limitation can be resolved by considering *observation histories*, and combining METRA with world models that use recurrent structures (e.g., Dreamer) would be an exciting future research direction. We have clarified this limitation in the revised manuscript (Appendix A).
>
> We thank the reviewer again for the helpful feedback and please let us know if there are any additional concerns or questions.
>
> [1] Eysenbach et al., Diversity is all you need: Learning skills without a reward function (2019).
>
> [2] Sharma et al., Dynamics-aware unsupervised discovery of skills (2020).
>
> [3] Park et al., Lipschitz-constrained unsupervised skill discovery (2022).
>
> [4] Laskin et al., Unsupervised reinforcement learning with contrastive intrinsic control (2022).

---

> > ### Author Response · Authors · 2023-11-21
> > **Gentle Reminder for Reviewer Feedback**
> >
> > We greatly appreciate your time and dedication to providing us with your valuable feedback. If there is anything else that needs clarification or further discussion, please do not hesitate to let us know.

---

### Official Review · Reviewer_1oDh · 2023-10-30

**Soundness:** 3 good
**Presentation:** 3 good
**Contribution:** 3 good
**Rating:** 8
**Confidence:** 5

**Summary:**

The paper presents a novel unsupervised reinforcement learning (RL) method, Metric-Aware Abstraction (METRA), which aims to make unsupervised RL scalable to complex, high-dimensional environments. The authors propose a new unsupervised RL objective that encourages an agent to explore its environment and learn a breadth of potentially useful behaviors without any supervision. The key idea is to cover a compact latent space that is metrically connected to the state space by temporal distances, instead of directly covering the state space. The authors demonstrate that METRA can discover a variety of useful behaviors in complex environments, outperforming previous unsupervised RL methods.

I have read the response and the authors address my concerns. I have raised my rating to accept.

**Strengths:**

1. The paper introduces a novel unsupervised RL objective, METRA, which is a significant contribution to the field. The idea of using temporal distances as a metric for the latent space is innovative and provides a new perspective on unsupervised RL.
2. The paper is technically sound, and the proposed method is well-motivated and clearly explained. The authors provide a thorough theoretical analysis of their method, including a connection to principal component analysis (PCA).
3. The paper is well-written and organized. The authors do a good job of explaining the motivation behind their method, the details of the method itself, and the experimental setup.

**Weaknesses:**

1. While the paper presents results on a variety of environments, it would be beneficial to see how METRA performs on more complex environments such as Atari[1] or Google Research Football[2]. This would provide a more comprehensive evaluation of the method's scalability and effectiveness.
2. The paper could benefit from a comparison with more diversity RL baselines, such as RSPO[3] and DGPO[4]. This would provide a more complete picture of how METRA compares to other state-of-the-art methods in the field.

- [1] MG Bellemare, Y Naddaf, J Veness, and M Bowling. “The arcade learning environment: An evaluation platform for general agents.” Journal of Artificial Intelligence Research (2012).
- [2] Kurach, Karol, et al. "Google research football: A novel reinforcement learning environment." Proceedings of the AAAI conference on artificial intelligence. Vol. 34. No. 04. 2020.
- [3] Zhou, Zihan, et al. "Continuously discovering novel strategies via reward-switching policy optimization." arXiv preprint arXiv:2204.02246 (2022).
- [4] Chen, Wenze, et al. "DGPO: Discovering Multiple Strategies with Diversity-Guided Policy Optimization." arXiv preprint arXiv:2207.05631 (2022).

**Questions:**

1. How does METRA handle environments with non-stationary dynamics or environments where the temporal distance between states can change over time?
2. How does the dimensionality of the latent space affect the performance of METRA? Is there a trade-off between the dimensionality of the latent space and the complexity of the behaviors that can be learned?

---

> ### Author Response · Authors · 2023-11-14
>
> We thank the reviewer for the thorough review and constructive feedback about this work. Below, we describe how we have added new comparisons with DGPO and a new ablation study with different sizes of latent states. We believe that these changes strengthen the paper, and welcome additional suggestions for further improving the work.
>
> * **Additional baseline – DGPO**
>
> Thank you for suggesting the new baselines. We additionally compared METRA with DGPO [2] on the MuJoCo Ant and HalfCheetah environments. With the addition of this comparison, the paper compares to a total of $12$ different methods.
>
> Since DGPO [2] (as well as RSPO [1]) was originally proposed for the quality diversity (QD) setting (i.e., learn diverse behaviors *while maximizing rewards*), unlike our unsupervised RL setting (i.e., learn diverse behaviors *without rewards*), we only use the intrinsic motivation part of DGPO for our experiments. Specifically, we use $\min_{z’ \neq z} \log{\frac{q(z|s’)}{q(z|s’) + q(z’|s’)}}$ as the reward function for DGPO.
>
> | Environment (Metric) | DIAYN | DGPO | METRA (ours) |
> |---|---|---|---|
> | HalfCheetah (policy state coverage) | $6.75 \pm 2.22$ | $6.75 \pm 2.06$ | $\mathbf{186.75} \pm 16.21$ |
> | HalfCheetah (total state coverage) | $19.50 \pm 3.87$ | $22.25 \pm 5.85$ | $\mathbf{177.75} \pm 17.10$ |
> | Ant (policy state coverage) | $11.25 \pm 5.44$ | $7.00 \pm 3.83$ | $\mathbf{1387.75} \pm 77.38$ |
> | Ant (total state coverage) | $107.75 \pm 17.00$ | $121.50 \pm 4.36$ | $\mathbf{6313.25} \pm 747.92$ |
>
> The table above compares the performances of METRA, DIAYN, and DGPO on Ant and HalfCheetah (at epoch 10000, 4 seeds each). The results suggest that both DIAYN and DGPO struggle to cover the state space in the absence of supervision. This is an expected result, given that DGPO also considers a **KL divergence** as a divergence measure, which is a *metric-agnostic* quantity. As such, DGPO shares the same limitation as DIAYN of not necessarily encouraging exploration (Section 2). Hence, while DGPO succeeds in the QD setting with the guidance of rewards, it does not necessarily achieve strong performance in our unsupervised settings.
>
> That being said, we believe combining existing KL-based intrinsic rewards, such as DGPO, with our temporal distance-based **Wasserstein** metric could potentially lead to another performant unsupervised RL algorithm, which we leave for future work. We have updated the new results in the draft (Appendix F.4).
>
> * **How does the dimensionality of the latent space affect the performance of METRA? Is there a trade-off between the dimensionality of the latent space and the complexity of the behaviors that can be learned?**
>
> As the reviewer pointed out, the size of the latent skill space affects the performance of METRA. Since METRA maximizes state coverage *under the capacity of the latent space $\mathcal{Z}$*, a too small skill latent space could lead to less diverse skills. To empirically demonstrate this, we evaluate METRA with 1-D, 2-D, and 4-D continuous skills as well as 2, 4, 8, 16, and 24 discrete skills on MuJoCo Ant and HalfCheetah.
>
> | Environment | \| | 1-D skills | 2-D skills | 4-D skills | \| | 2 skills | 4 skills | 8 skills | 16 skills | 24 skills |
> |---|---|---|---|---|---|---|---|---|---|---|
> | HalfCheetah | \| | $-17.84 \pm 4.09$ | $-12.86 \pm 1.51$ | $\mathbf{-11.16} \pm 0.89$ | \| | $-26.76 \pm 12.36$ | $-10.85 \pm 1.80$ | $-11.43 \pm 0.63$ | $\mathbf{-9.97} \pm 1.52$ | $-12.10 \pm 1.75$ |
> | Ant | \| | $-29.11 \pm 6.35$ | $-7.58 \pm 3.90$ | $\mathbf{-7.54} \pm 0.90$ | \| | $-23.42 \pm 4.00$ | $-13.81 \pm 3.69$ | $-10.64 \pm 4.25$ | $-8.96 \pm 1.78$ | $\mathbf{-7.28} \pm 1.71$ |
>
> The table above shows the goal-reaching performances (negative goal distance, the higher the better) of different skill latent spaces (at epoch 10000, 4 seeds each). In general, the diversity of skill (and thus goal-reaching performance) increases as the capacity of $\mathcal{Z}$ grows. However, using a too large skill space may slow down the training of the skill policy, as the agent needs to learn more behaviors. We have added qualitative results from these experiments to the revised draft (Appendix F.3).

---

> > ### Author Response · Authors · 2023-11-14
> >
> > * **Applying METRA to other types of environments such as Atari or Google Research Football**
> >
> > Thank you for the suggestion. While we agree that evaluating METRA on more diverse benchmarks like Atari games would further strengthen the paper, we would like to note that we have included a large fraction of the most widely used benchmarks for unsupervised RL and unsupervised skill discovery [3, 4, 5, 6, 7, 8, 9, 10, 11, 12, 13], including both locomotion and manipulation environments. Please understand that adapting the current METRA implementation to Atari games or Football (with discrete action spaces) requires a considerable amount of time and work, as they require a very different RL backbone (other than SAC) due to the difference between continuous vs. discrete action spaces. (We also tried to find an existing unsupervised skill discovery implementation for Atari, but most prior methods do not evaluate on Atari games, and the only available [repository](https://github.com/mklissa/dceo) we found seems to require 5-7 days of training.) That being said, we believe further extending the idea behind METRA to other types of environments is an exciting future research direction.
> >
> > * **“How does METRA handle environments with non-stationary dynamics or environments where the temporal distance between states can change over time?”**
> >
> > METRA in its current form assumes a fixed MDP (i.e., stationary dynamics), as commonly assumed in many works in RL. We leave devising an unsupervised RL algorithm for non-stationary environments and continual learning settings for future work. We believe one potential way to deal with this issue is to combine METRA with recurrent latent state spaces [14]. We have clarified this limitation in the updated draft (Appendix A).
> >
> >
> > We thank the reviewer again for the helpful feedback and please let us know if there are any additional concerns or questions.
> >
> > [1] Zhou et al. Continuously discovering novel strategies via reward-switching policy optimization (2022).
> >
> > [2] Chen et al. Dgpo: Discovering multiple strategies with diversity-guided policy optimization (2022).
> >
> > [3] Eysenbach et al., Diversity is all you need: Learning skills without a reward function (2019).
> >
> > [4] Sharma et al., Dynamics-aware unsupervised discovery of skills (2020).
> >
> > [5] Mendonca et al., Discovering and achieving goals via world models (2021).
> >
> > [6] Laskin et al., Urlb: Unsupervised reinforcement learning benchmark (2021).
> >
> > [7] He et al., Wasserstein unsupervised reinforcement learning (2022).
> >
> > [8] Park et al., Lipschitz-constrained unsupervised skill discovery (2022).
> >
> > [9] Zhao et al., A mixture of surprises for unsupervised reinforcement learning (2022).
> >
> > [10] Shafiullah and Pinto, One after another: Learning incremental skills for a changing world (2022).
> >
> > [11] Laskin et al., Unsupervised reinforcement learning with contrastive intrinsic control (2022).
> >
> > [12] Park et al., Controllability-aware unsupervised skill discovery (2023).
> >
> > [13] Yang et al., Behavior contrastive learning for unsupervised skill discovery (2023).
> >
> > [14] Xie et al., Deep reinforcement learning amidst lifelong non-stationarity (2020).

---

### Official Review · Reviewer_CufN · 2023-10-31

**Soundness:** 4 excellent
**Presentation:** 4 excellent
**Contribution:** 4 excellent
**Rating:** 8
**Confidence:** 4

**Summary:**

This paper introduces a novel unsupervised RL objective called Metric-Aware Abstraction (METRA). The objective is to only learn to explore on a compact latent space which is metrically connected to the state space by temporal distances. The learned skills on this latent space are scalable to a variety of downstream control tasks. METRA is the first unsupervised RL method that demonstrates the discovery of diverse locomotion behaviors in pixel-based tasks.

**Strengths:**

- The empirical study of this paper is very sound and solid. The paper evaluates the method on various control tasks, including locomotion and manipulation tasks. Besides, the paper aims to address the unsupervised RL problem on visual-based tasks, which are much more challenging in the area. The paper also compare the results to multiple previous works, showing the significant improvement on skill discovery.

- The methodology part is very organized. The authors aim to maximize state converage under a specific metrics, which should be scalable to pixel-based tasks. Then temporal distance makes sense and is easy to be turned to an constrainted optimization problem.

- The paper is very well-written. The background in Section 2 and 3 clearly shows the motivation of this work and the connections to the previous methods. The method and empirical study both illustrate many details and the source code is linked, which make this work easy to understand and follow.

**Weaknesses:**

- The paper can be more impactful and solid if the method is deployed on the real world tasks, like locomotion control on a real robot. Besides, as the authors have already listed in Appendix A, the method can be combined to more recent RL works.

**Questions:**

- In Figure 8, the LEXA and DIAYN totally failed to handle most of the tasks (especially DIAYN). Is this because the skill discovery process has already failed or the learned skills is useless on downstream tasks?

- DIAYN, DADS, and other previous works only consider state space tasks, so the objectives of them are not able to train the visual encoder. But the METRA objective is much more suitable to learn a visual representation. Can the authors analyze more on the advantage of the proposed method on the visual representation learning? If the vision encoders of all baselines are the same (a pretrained network), will the experiment results change?

- Similar to the last question, will the proposed method still outperform if we only consider state-based tasks? Does the advantage come from visual learning?

- In Figure 7, some methods are very unstable on the Kitchen tasks (variance of different seeds is large). Can the authors give any reason?

---

> ### Author Response · Authors · 2023-11-14
>
> We thank the reviewer for the thorough review and constructive feedback about this work. Please find our answers to the questions below. We welcome additional suggestions for further improving the work.
>
> * **Failure of DIAYN/LEXA in goal-conditioned tasks**
>
> As the reviewer pointed out, DIAYN and LEXA struggle with complex goal-conditioned tasks. This is mainly because they often fail to cover the state space, as shown in Figure 3 (for LEXA, please refer to the “P2E/Disag.” column, which is the exploration method LEXA uses). This is an expected result, given that DIAYN maximizes mutual information, which does not necessarily encourage exploration (Section 2), and LEXA (a pure exploration method) struggles in complex environments, where it is infeasible to completely cover every possible state or fully capture environment dynamics.
>
> * **Does the advantage of METRA mainly come from visual learning?**, **“Will the proposed method still outperform if we only consider state-based tasks?”**
>
> In our experiments, Ant and HalfCheetah are state-based environments (Fig. 4), in which METRA outperforms most previous unsupervised RL methods by significant margins (Fig. 5, Fig. 7). This suggests that the performance gain mainly comes from our objective, not from visual representation learning. In state-based environments, however, LSD (a previous Euclidean-distance maximizing skill discovery method) shows comparable performance to METRA. Yet, LSD fails to scale to pixel-based environments because Euclidean distances do not necessarily provide meaningful learning signals in pixels.
>
> The fact that the performance gain of METRA mainly comes from its objective becomes further clearer if we contrast the objectives of DIAYN and METRA: DIAYN essentially minimizes $\\\|\phi(s) - z\\\|_2^2$ (Appendix E.1), while METRA maximizes $(\phi(s') - \phi(s))^\top z$ under the $\\\|\phi(s) - \phi(s')\\\|_2 \leq 1$ constraint (Eq. 8). They both have the exactly same (visual) encoder $\phi$ in their objectives, but the results are very different (Fig. 3). This is, again, because mutual information does not encourage the agent to cover the state space (Section 2).
>
> * **“In Figure 7, some methods are very unstable on the Kitchen tasks (variance of different seeds is large). Can the authors give any reason?”**
>
> We believe this is mainly because we use a *queue* (or *policy*) coverage metric for Kitchen, while we use a *total* coverage metric for the other environments (please refer to Appendix G.1 for details). The rationale behind this choice is that, as mentioned in Sec 5.3, most methods max out the total coverage metric in Kitchen. The queue coverage metric in Kitchen measures the number of achieved subtasks from the trajectories within a specific window. Exploration methods may exhibit high variances in this metric since they may visit different parts of the space on different iterations. That being said, we would like to note that METRA outperforms the best exploration method for Kitchen (P2E = LEXA) in a different goal-reaching metric by a statistically significant margin (Fig. 8).
>
> We thank the reviewer again for the helpful feedback and please let us know if there are any additional concerns or questions.

---

> > ### Comment · Reviewer_CufN · 2023-11-14
> > **Reply to the authors**
> >
> > Thank you for the considerate comments. I believe the given answers have addressed all my concerns.

---

### Meta-Review · Area_Chair_CWNj · 2023-12-06

**Metareview:**

METRA is a novel unsupervised RL objective that learns to explore on a compact latent space which is metrically connected to the original state space by temporal distances. The paper provides a through theoretical analysis of the proposed technique. Empirical results show that the exploration technique is able to learn diverse behaviors in high dimensional environments. Some additional experimental comparisons have been suggested by the reviewers that would make the paper more impactful.

**Justification For Why Not Higher Score:**

N/A

**Justification For Why Not Lower Score:**

The paper introduces a novel unsupervised RL objective, provides theoretical motivation and justification for the objective, and shows it works in a diverse set of RL environments. There do not seem to be any significant shortcomings in the paper.

---

### Decision · Program_Chairs · 2024-01-16

Accept (oral)